# Anti-infective therapy using species-specific activators of *Staphylococcus aureus* ClpP

Bingyan Wei [1,2,3,7], Tao Zhang [2,7], Pengyu Wang [1,2,3,7], Yihui Pan [1,2,3], Jiahui Li [2,3], Weizhong Chen [4], Min Zhang [5], Quanjiang Ji [4], Wenjuan Wu [5], Lefu Lan [1,2,3], Jianhua Gan [6] & Cai-Guang Yang [1,2,3] ✉

The emergence of methicillin-resistant *Staphylococcus aureus* isolates highlights the urgent need to develop more antibiotics. ClpP is a highly conserved protease regulated by ATPases in bacteria and in mitochondria. Aberrant activation of bacterial ClpP is an alternative method of discovering antibiotics, while it remains difficult to develop selective *Staphylococcus aureus* ClpP activators that can avoid disturbing *Homo sapiens* ClpP functions. Here, we use a structure-based design to identify (*R*)- and (*S*)-ZG197 as highly selective *Staphylococcus aureus* ClpP activators. The key structural elements in *Homo sapiens* ClpP, particularly W146 and its joint action with the C-terminal motif, significantly contribute to the discrimination of the activators. Our selective activators display wide antibiotic properties towards an array of multidrug-resistant staphylococcal strains in vitro, and demonstrate promising antibiotic efficacy in zebrafish and murine skin infection models. Our findings indicate that the species-specific activators of *Staphylococcus aureus* ClpP are exciting therapeutic agents to treat staphylococcal infections.

*Staphylococcus aureus* (*S. aureus*) colonizes the human skin and nares and frequently causes invasive and life-threatening diseases[1]. Methicillin-resistant *S. aureus* (MRSA) is responsible for many hospital-acquired infections, which include surgical site infections, bacteremia, and sepsis[2]. MRSA has acquired multidrug resistance against many antibiotics, including β-lactam, cephalosporin, fluoroquinolone, aminoglycoside, tetracycline, macrolide, and trimethoprim-sulfamethoxazole[3]. Due to this resistance, antibiotic prophylaxis of MRSA infections frequently incites nosocomial diseases such as *Clostridium difficile* infection[4]. Given the dwindling supply of potent antibiotics, it is increasingly important to develop new methods of treating multidrug-resistant MRSA infections[5]. To this end, developing antistaphylococcal agents, particularly those with a different mode of action, is urgently needed to provide future treatments for MRSA infections[6–11].

Caseinolytic protease P (ClpP) is a highly conserved protease regulated by ATPases in bacteria and in human mitochondria, and plays an important role in protein quality control by regulating and degrading the misfolded or damaged proteins[12,13]. In bacteria, ClpP is responsible for modulating the expression of virulence factors, antibiotic resistance, and the formation of biofilms and persisters[14–17]. Likewise, *Homo sapiens* ClpP (*Hs*ClpP), which is found in the mitochondria, is a key protease regulating mitochondrial proteins homeostasis. These proteins are mainly involved in respiratory chain complexes[18]. The dysregulated expression or tissue distribution of *Hs*ClpP is strongly associated with diseases, such as cancer and neurological disorders[19,20].

The dysfunction of ClpP by small molecule activators has recently been demonstrated to be a possible method of discovering

[1]School of Pharmaceutical Science and Technology, Hangzhou Institute for Advanced Study, University of Chinese Academy of Sciences, Hangzhou 310024, China. [2]State Key Laboratory of Drug Research, Centre for Chemical Biology, Shanghai Institute of Materia Medica, Chinese Academy of Sciences, Shanghai 201203, China. [3]University of Chinese Academy of Sciences, Beijing 100049, China. [4]School of Physical Science and Technology, ShanghaiTech University, Shanghai 201210, China. [5]Department of Laboratory Medicine, Shanghai East Hospital, Tongji University School of Medicine, Shanghai 200123, China. [6]School of Life Sciences, Fudan University, Shanghai 200433, China. [7]These authors contributed equally: Bingyan Wei, Tao Zhang, Pengyu Wang. ✉e-mail: yangcg@simm.ac.cn

antibacterial and anticancer drugs[21]. Remarkably, acyldepsipeptides (ADEPs) has been identified as a promising class of antibiotics, which reveal that bacterial ClpP is a possible antibiotic drug target[22,23]. These naturally occurring activators allosterically switch *S. aureus* ClpP (*Sa*ClpP) to a gain-of-function state to degrade multiple essential proteins, such as the filamentous temperature-sensitive Z (*Sa*FtsZ), which inhibits cell division and eventually leads to bacterial cells death[24,25]. In addition, ADEP 4 suppresses the growth of persisters and eradicates a chronic biofilm infection when combined with marketed antibiotics[23], indicating that the therapeutic application of ADEPs can be used in combination therapy. However, the chemical instability issues of ADEP analogs could significantly restrict the applications of antibiotic drug discovery[26]. The discovery of ADEPs as bacterial ClpP activators has inspired studies on *Hs*ClpP. The aberrant activation of *Hs*ClpP protease activity significantly degrades multiple proteins in the respiratory chain complex, which exerts antitumor effects[27]. Of note, ADEP-28, an analog of ADEP 4, was shown to activate *Hs*ClpP and induce cytotoxic effects by activating intrinsic, caspase-dependent apoptosis[28]. Additionally, imipridone ONC201 and ONC212 activate the *Hs*ClpP protease and facilitate ClpP-mediated proteolysis, resulting in cancer cell lethality[27,29]. ONC212 also functions as an antibiotic, which activates ClpPs from *Escherichia coli*, *Bacillus subtilis*, and *S. aureus* in biochemical and genetic assays and eventually suppressed bacterial cell proliferation[30]. Recently, we found that ONC212 and ADEP 4 dysregulated *Xoo*ClpP to degrade *Xoo*FtsZ, which suggests a promising strategy of treating leaf blight diseases[31]. Other activators, such as bengamide analogs and activators of self-compartmentalizing proteases (ACPs), also exhibit certain activities on ClpP[32–34]. However, to date, there is no species-specific activator for the *Sa*ClpP protease.

In this study, we report on two highly selective *Sa*ClpP activators, (*R*)- and (*S*)-ZG197, which selectively activates *Sa*ClpP rather than *Hs*ClpP. Additionally, (*R*)-ZG197 exhibits a broad range of antibiotic properties towards an array of MRSA strains and shows promising antibiotic efficacy in vivo.

## Results

### Identification of ZG180 as a *Sa*ClpP activator

Since ADEP 4 and ONC212 are global activators for ClpP proteases (Supplementary Fig. 1a), our goal is to develop small molecule activators that selectively enhance the *Sa*ClpP protease activity over *Hs*ClpP (Fig. 1a). Given the established structure-activity relationships of ADEP 4 and ONC212 as ClpP activators[29,35–38], we search for distinct chemical scaffolds as selective activators for *Sa*ClpP. To this end, we conducted a high-throughput screening (HTS) on a compound library containing 3,896 candidates. ICG-001 was confirmed to promote hydrolysis of the fluorescein isothiocyanate-labelled casein (FITC-casein) in the presence of *Sa*ClpP (Supplementary Fig. 1b and 1c). ICG-001 was initially characterized as a Wnt signaling pathway inhibitor, while it was recently screened as an activator for *Hs*ClpP[39,40]. Considering distinct scaffold of ICG-001 and the practicable synthesis of derivatives, we chose ICG-001 as a starting point for follow-up synthetic optimization[41]. Incorporation with the isoleucine fragment and 4,4,4-trifluorobutyl motif into the core scaffold of ICG-001 provided a potent ClpP activator ZG180 (Fig. 1b). ZG180 is more potent than ICG-001 in activating *Sa*ClpP for α-casein hydrolysis, as evidenced by a 3-fold decrease in EC$_{50}$ (Fig. 1c and Supplementary Fig. 1d). However, ZG180 also promotes *Hs*ClpP for α-casein hydrolysis (Fig. 1c and Supplementary Fig. 1e), which indicates that ZG180 is also a global activator for both bacterial and mitochondrial ClpP proteins.

To investigate the molecular mechanism of our activator-promoted proteolysis of ClpP proteases, we resolved the crystal structure of ZG180 bound to *Sa*ClpP (7WID in PDB) and *Hs*ClpP (7WH5 in PDB) to a resolution of 1.90 Å and 2.13 Å, respectively (Supplementary Table 1). Either ZG180/*Sa*ClpP or ZG180/*Hs*ClpP structure is a single tetradecamer composed of two stacked heptameric rings

(Fig. 1d, e). Like the previously characterized ClpP activators, the binding of fourteen ZG180 molecules induces an open-gated state in the tetradecameric ClpP cylinders with an enlarged axis entrance pore[42–45]. The apparent electron densities in the 2fo-fc and fo-fc omit maps unambiguously demonstrate ZG180 binding on the apical surface of *Sa*ClpP and *Hs*ClpP (Fig. 1f, g). The naphthyl motif and trifluorobutyl chain in ZG180 form hydrophobic stacking in the binding pockets of *Sa*ClpP, which contributes to the enhanced interactions. A network of hydrogen bonds between ZG180 and the side chains of D27, L49, Q52, Y61, and Q89 collectively increase the binding affinity to *Sa*ClpP (Fig. 1f). A similar binding manner is observed in the ZG180/*Hs*ClpP structure (Fig. 1g). These structural features are similar to the previously reported structural complexes of both *Sa*ClpP and *Hs*ClpP bound with small molecule activators, such as ADEP 4/*Sa*ClpP, ADEP-28/*Hs*ClpP, and ONC201/*Hs*ClpP[27,28,35], in which all the activator molecules are similarly positioned in the hydrophobic pockets and prevent ATPase binding to ClpP.

### Development of (*R*)- and (*S*)-ZG197 as selective *Sa*ClpP activators by structure-based design

To design a specific activator for *Sa*ClpP over *Hs*ClpP, we analyzed the amino acid sequences that can affect the binding mode of ZG180 in two ClpP proteins. While *Sa*ClpP and *Hs*ClpP share a high similarity in the primary sequences (Supplementary Fig. 2a), there is a large indole motif of W146 in *Hs*ClpP but a much smaller isopropyl side chain of I91 in *Sa*ClpP in the conserved position. Indeed, structural superimposition of the ZG180/*Sa*ClpP and ZG180/*Hs*ClpP complexes also reveals that the α-carbon of the naphthyl motif in ZG180 is likely to be a discriminant site for introducing steric clash against the bulky side chain of W146 in *Hs*ClpP (Fig. 2a). To this end, we incorporated a methyl group on ZG180 to obtain two naphthalen-1-ylmethyl analogs (*R*)- and (*S*)-ZG197 (Fig. 2b). Similar to ZG180, both (*R*)-ZG197 and (*S*)-ZG197 activate *Sa*ClpP to degrade α-casein and *Sa*FtsZ in vitro (Fig. 2c, Supplementary Fig. 2b, c). Unlike the nonselective ZG180 activator, (*R*)-ZG197 activates *Sa*ClpP with an EC$_{50}$ value of 1.5 μM that is a 20-fold higher activity than that activates *Hs*ClpP, while (*S*)-ZG197 fails to activate *Hs*ClpP even at 100 μM (Fig. 2c and Supplementary Fig. 2d). Moreover, both (*R*)-ZG197 and (*S*)-ZG197 promote the degradation of cellular *Sa*FtsZ protein in cell lysates of *S. aureus* 8325-4 *clpP* knockout (Δ*clpP*) strain with the supplement of the recombinant *Sa*ClpP protein, but no *Sa*FtsZ degradation occurred in the presence of the recombinant *Hs*ClpP protein. In contrast, the global activator ONC212 activates both *Sa*ClpP and *Hs*ClpP to degrade the cellular *Sa*FtsZ protein (Fig. 2d). In conclusion, by using structure-based design, we successfully developed the species-specific *Sa*ClpP activators (*R*)- and (*S*)-ZG197 that minimally disturb the activity of mitochondrial ClpP in vitro.

### Characterization on the interactions between our activators and *Sa*ClpP

To explain why our activators selectively activate *Sa*ClpP rather than *Hs*ClpP in vitro, we measured the direct interactions between (*R*)- or (*S*)-ZG197 compounds and the two ClpP proteases. During the differential scanning fluorimetry (DSF) analysis, (*R*)-ZG197 significantly enhances the thermal stability of *Sa*ClpP in a dose-dependent manner while having a weak effect on *Hs*ClpP (Fig. 3a). This indicates that (*R*)-ZG197 exhibited a stronger binding affinity for *Sa*ClpP than *Hs*ClpP. Similarly, (*S*)-ZG197 increases the melting temperature (T$_m$) of *Sa*ClpP but barely changes the T$_m$ of *Hs*ClpP (Fig. 3b). We then conducted an isothermal titration calorimetry (ITC) assay to determine the stoichiometries of the binding capacity. The dissociation constant (K$_d$) for (*R*)- and (*S*)-ZG197 binding to *Sa*ClpP is quantitatively estimated to be 2.5 ± 0.2 μM and 5.0 ± 0.3 μM, respectively (Fig. 3c). We also analyzed our activators' binding to *Sa*ClpP and *Hs*ClpP in biolayer interferometry (BLI). Likewise, (*R*)- and (*S*)-ZG197 display a K$_d$ of 58 ± 4.8 nM

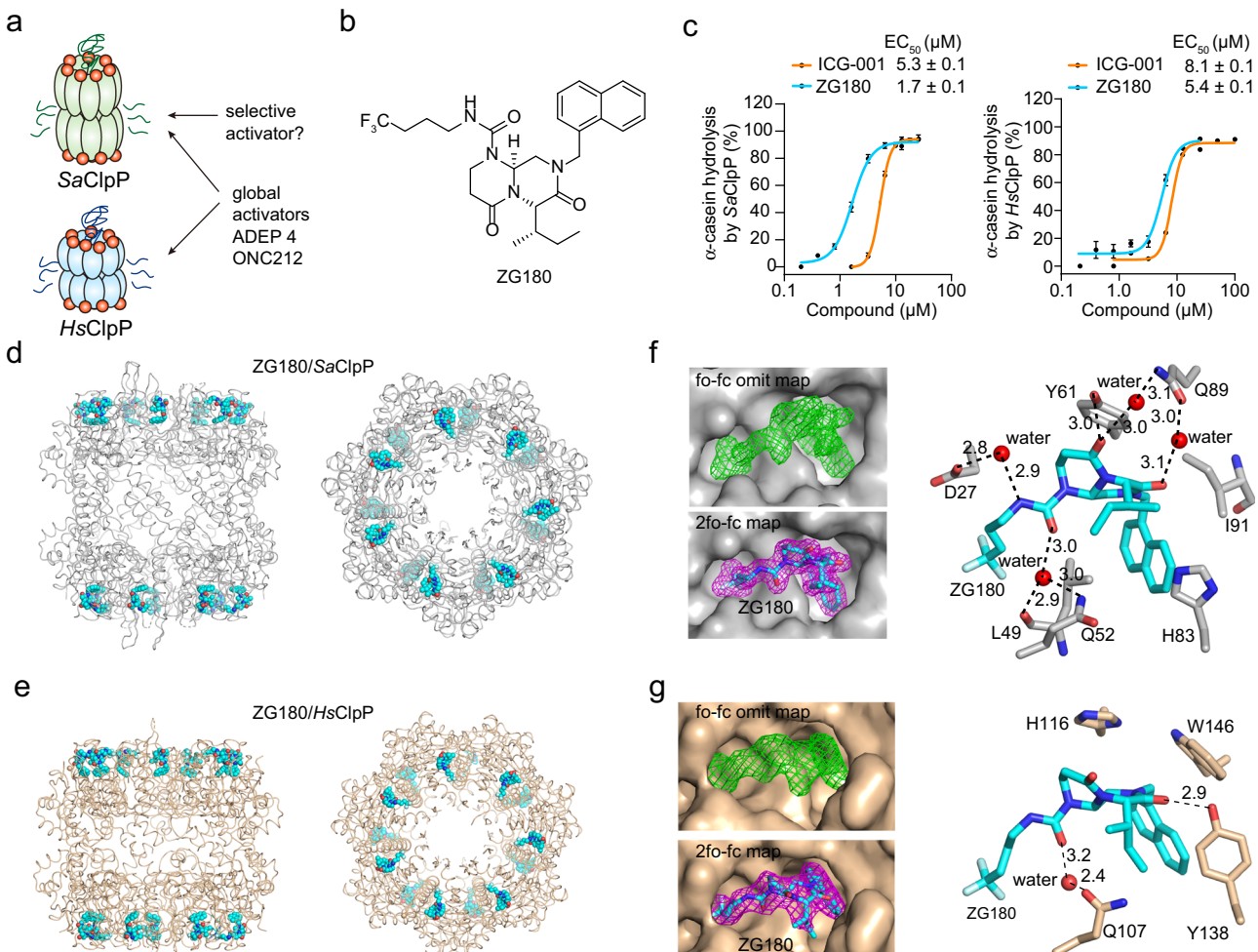

**Fig. 1 | Identification of ZG180 as potent ClpP activator. a** The goal of this work is to develop selective *Sa*ClpP activators. **b** The synthetic lead compound ZG180 as ClpP activator. **c** Quantitation of α-casein hydrolysis by *Sa*ClpP and *Hs*ClpP in the presence of ICG-001 and ZG180, respectively (*n* = 3 biologically independent experiments). Data are presented as mean ± SD (error bars). X-ray crystal structure of ZG180/*Sa*ClpP (**d**) and ZG180/*Hs*ClpP (**e**) complexes. Side and top views of the structural complexes are displayed. *Sa*ClpP is shown in gray cartoon tube, *Hs*ClpP is

shown in wheat cartoon tube, and ZG180 is shown in cyan sphere. **f, g** The electron density maps and close view of ZG180 binding in the hydrophobic pockets of *Sa*ClpP (**f**) and *Hs*ClpP (**g**). The fo-fc omit map is contoured at 3.0 Å and colored in green, while the 2fo-fc map is contoured at 1.0 Å and colored in magenta. ZG180 is shown in cyan stick. Hydrogen bonds are indicated with dark dashed lines, and the distance in Å is labeled. Source data are provided as a Source Data file.

and 470 ± 60 nM for binding to *Sa*ClpP, respectively (Supplementary Fig. 3a). However, neither (*R*)-ZG197 nor (*S*)-ZG197 exhibits detectable binding affinity to *Hs*ClpP in BLI analysis (Supplementary Fig. 3b). This indicates that our activators selectively bind to *Sa*ClpP rather than *Hs*ClpP in vitro.

We next investigated whether our activators directly interact to *Sa*ClpP in staphylococcal cells by performing the cellular thermal shift assay (CETSA). We observed significantly increased thermal stability of *Sa*ClpP when the *S. aureus* 8325-4 strain was cultured with 10 μM (*R*)- or (*S*)-ZG197 (Fig. 3d and Supplementary Fig. 3c). Similar elevation of *Sa*ClpP $T_m$ occurred when bacterial cells were treated with ONC212. This indicates that our selective activators and the global activator ONC212 can directly bind to the *Sa*ClpP protease in staphylococcal cells. We also performed CETSA in HEK 293T/17 cell line to test whether our activators bind to *Hs*ClpP. Similar to DMSO, (*R*)- and (*S*)-ZG197 have minimal effects on the thermal stability of *Hs*ClpP in HEK 293 T/17 cells, while ONC212 significantly increased *Hs*ClpP stability (Fig. 3e and Supplementary Fig. 3d). Therefore, the results in cellular binding assay highlight direct and selective interactions of our activators towards *Sa*ClpP but not *Hs*ClpP in cellular contexts, which is similar to the results from the in vitro biophysical binding assays.

## Identification of critical residues/motif responsible for our activators' selective binding and activation of *Sa*ClpP but not *Hs*ClpP

To reveal the structural insight into the mechanism of (*R*)- and (*S*)-ZG197' selective activation on *Sa*ClpP proteolysis, we determined the X-ray crystal structures of (*R*)-ZG197/*Sa*ClpP (7XBZ in PDB) and (*S*)-ZG197/*Sa*ClpP (7WGS in PDB) complexes to a resolution of 2.15 Å and 2.11 Å, respectively (Supplementary Table 1). The resolved structures demonstrate that twelve (*R*)-ZG197 or fourteen (*S*)-ZG197 activators bind in the fourteen hydrophobic pockets of *Sa*ClpP (Supplementary Fig. 4a), and the electron density maps distinctly show the existence of (*R*)- and (*S*)-ZG197, respectively (Fig. 4a, b). The naphthyl motifs of (*R*)- and (*S*)-ZG197 point in different directions due to the chiral methyl substituents. The carbonyl group in both (*R*)- and (*S*)-ZG197 forms a hydrogen bond with H83 of *Sa*ClpP, and the side chain of Q52 also forms a hydrogen bond with (*R*)-ZG197 directly or (*S*)-ZG197 mediated by a water molecule. Additional hydrogen bonds mediated by the side chains of D27, L49 jointly contribute to (*S*)-ZG197 binding to *Sa*ClpP (Fig. 4a, b). In addition, extensive hydrophobic interactions are observed on (*R*)- and (*S*)-ZG197 in ligand binding sites of *Sa*ClpP (Supplementary Fig. 4b). Structural alignment of *Sa*ClpP (6TTY in

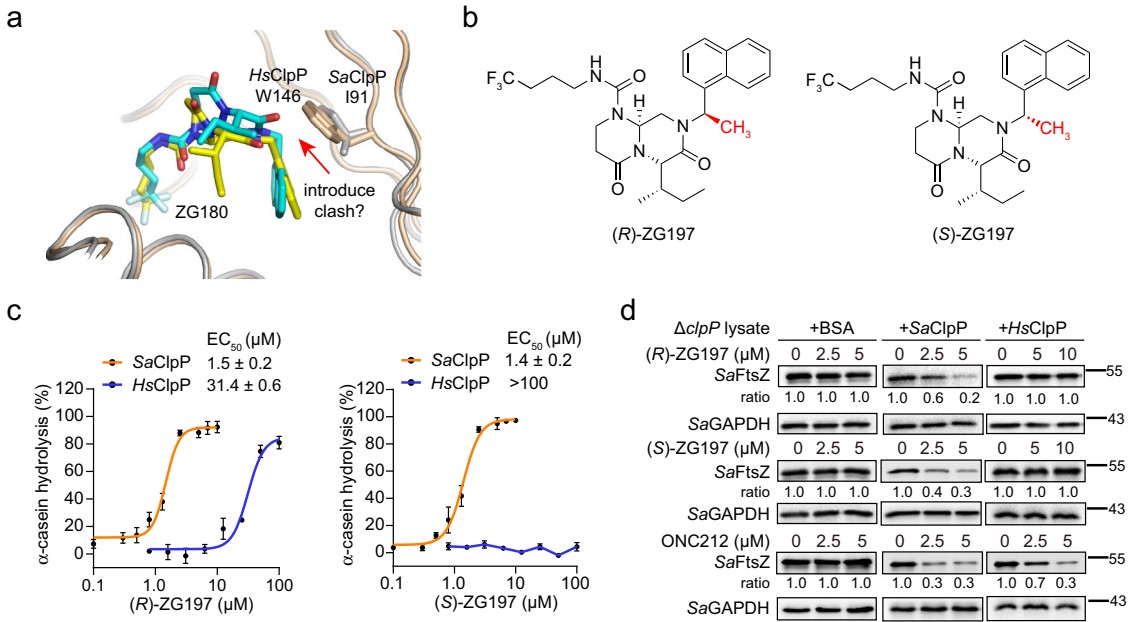

**Fig. 2 | Design and development of selective *Sa*ClpP activators. a** Structural superimposition of ZG180/*Sa*ClpP and ZG180/*Hs*ClpP complexes to reveal the strategy for designing selective activators for *Sa*ClpP over *Hs*ClpP. *Sa*ClpP and *Hs*ClpP are shown in gray and wheat cartoon tubes, respectively. ZG180 is colored in cyan in the ZG180/*Sa*ClpP structure, while it is colored in yellow in the ZG180/*Hs*ClpP structure. I91 in *Sa*ClpP and W146 in *Hs*ClpP are shown as a stick. **b** Structures of *Sa*ClpP activators (*R*)- and (*S*)-ZG197 developed by the structure-based design outlined in (**a**). The chiral methyl group is indicated in red. **c** Quantitation of α-casein hydrolysis by *Sa*ClpP and *Hs*ClpP in the presence of (*R*)-

and (*S*)-ZG197, respectively (*n* = 3 biologically independent experiments). Data are presented as mean ± SD (error bars). **d** Effect of activators on degradation of the cellular *Sa*FtsZ protein with the supplement of bovine serum albumin (BSA), the recombinant *Sa*ClpP, and *Hs*ClpP, respectively, in cell lysates of the *clpP* deletion (Δ*clpP*) mutant. The abundance of cellular *Sa*FtsZ is quantitated with a Western blot assay. Quantitation on the band intensity of *Sa*FtsZ is normalized against the loading control of *Sa*GAPDH, and the ratio in the DMSO group is considered as 1.0. Source data are provided as a Source Data file.

PDB), ADEP 4/*Sa*ClpP (6TTZ in PDB), (*R*)-ZG197/*Sa*ClpP (7XBZ in PDB) and (*S*)-ZG197/*Sa*ClpP (7WGS in PDB) yield low Cα root mean square deviation (RMSD) values around 0.2 Å for ClpP monomer, indicating a high similarity of ClpP folding (Supplementary Fig. 4c)[43,46,47]. Like the global activator ADEP 4, (*R*)- and (*S*)-ZG197 bindings induce an extended state of *Sa*ClpP, which has been characterized by a straight orientation of the α5 helix[26]. Moreover, the catalytic triad (S98, H123 and D172) are also displayed in highly similar orientations in these structures (Supplementary Fig. 4d). Interestingly, only four or five out of the fourteen N-terminal loops are ordered in the structural complexes of (*R*)- and (*S*)-ZG197/*Sa*ClpP, implying the highly dynamic nature of these motifs upon activators binding.

To understand the structural mechanism that causes (*R*)- and (*S*)-ZG197 to selectively bind to and activate *Sa*ClpP but not *Hs*ClpP, we aligned our resolved structures of (*R*)-ZG197/*Sa*ClpP and (*S*)-ZG197/*Sa*ClpP to the ZG180/*Hs*ClpP structure. The naphthyl group in (*S*)-ZG197 is positioned in the opposite direction compared to that in ZG180 and (*R*)-ZG197, which results in a direct clash with the side chain of W146 in *Hs*ClpP. This can restrict (*S*)-ZG197 to bind to *Hs*ClpP in a mode similar to *Sa*ClpP (Fig. 4c). While though the naphthyl group in (*R*)-ZG197 is positioned away from the side chain of W146 in *Hs*ClpP, the methyl substituent in (*R*)-ZG197 can generate a spatial hindrance and clash with W146 in *Hs*ClpP, which likely explain why (*R*)-ZG197 cannot interact with *Hs*ClpP. The structural alignment analysis suggests that I91 in *Sa*ClpP and W146 in *Hs*ClpP are primarily critical sites responsible for our activators' selective binding and activating on *Sa*ClpP but not *Hs*ClpP.

To illustrate how I91 influences our activators' binding to *Sa*ClpP, we constructed an I91W mutant in *Sa*ClpP to mimic the corresponding W146 in *Hs*ClpP and determined whether (*R*)- and (*S*)-ZG197 bind to and activate the *Sa*ClpPI91W mutant. As predicted, (*S*)-ZG197 exhibits a significantly diminished activity on the *Sa*ClpPI91W mutant for

α-casein hydrolysis, while (*R*)-ZG197 and ONC212 still activate *Sa*Clp-PI91W to degrade α-casein in vitro (Fig. 4d and Supplementary Fig. 4e). We next investigated the interactions between ClpP activators and the *Sa*ClpPI91W mutant in DSF. Consistent with the biochemical observation, (*S*)-ZG197 cannot thermally stabilize the *Sa*ClpPI91W mutant, while (*R*)-ZG197 and ONC212 still significantly increase the $T_m$ of the *Sa*ClpPI91W mutant (Fig. 4e). We then quantitatively determined the affinity of (*R*)-ZG197 for binding to *Sa*ClpPI91W, and the $K_d$ is 0.93 ± 0.2 μM in ITC assay. No binding was observed between (*S*)-ZG197 and *Sa*ClpPI91W (Supplementary Fig. 4f). It should be noticed that (*R*)-ZG197 is still positive on the *Sa*ClpPI91W mutant. This is because of the methyl substituent in (*R*)-ZG197 that can contribute hydrophobic affinity with W91 in the *Sa*ClpPI91W mutant, while it likely clashes with W146 in *Hs*ClpP (Fig. 4c). A resolved structure of (*R*)-ZG197/*Sa*ClpPI91W complex would be beneficial for revealing the mechanism of this phenomenon. In addition, we constructed the complemented *clpPI91W* mutant strain via a tetracycline-inducible expression vector pYJ335 in the Δ*clpP* mutant (Supplementary Fig. 4g), and assessed the cellular interactions between the activators and the *Sa*ClpPI91W mutant in the complemented strain. In line with the observed interactions in vitro, (*S*)-ZG197 treatment fails to induce the $T_m$ shift of *Sa*ClpPI91W in intact staphylococcal cells, while (*R*)-ZG197 and the global activator ONC212 exhibit strong effects on *Sa*ClpPI91W stabilization (Fig. 4f and Supplementary Fig. 4h).

We also constructed the *Hs*ClpPW146A mutant to test whether the proposed steric clash between our activators and the rigid side chain of W146 restricts their binding to *Hs*ClpP. In contrast to the observation that (*R*)-ZG197 is inactive on *Hs*ClpP, (*R*)-ZG197 becomes active to promote the *Hs*ClpPW146A mutant for α-casein hydrolysis (Fig. 4g and Supplementary Fig. 4i), and displays binding affinity towards *Hs*ClpPW146A as measured in DSF assay (Fig. 4h). ONC212 always binds to and activates both *Hs*ClpP and the W146A mutant in vitro. However,

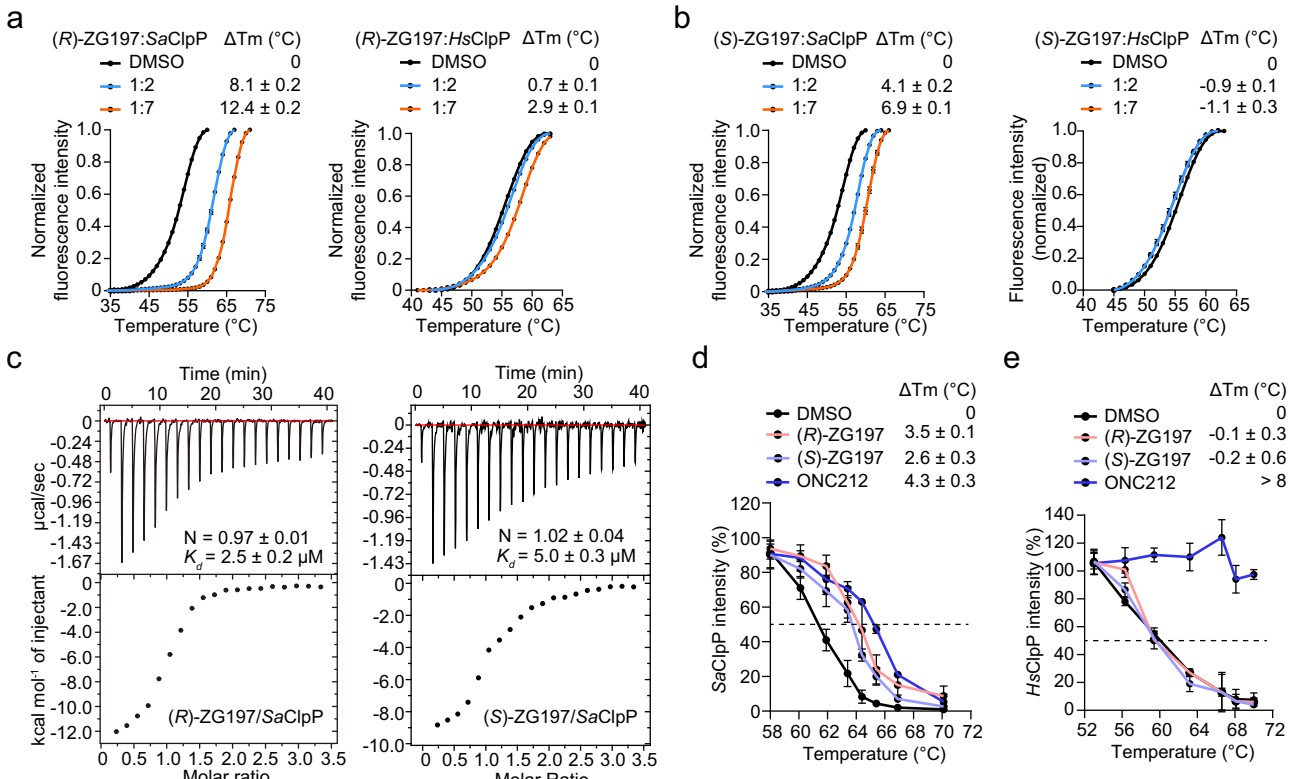

**Fig. 3 | Investigation of the interaction between species-selective activators and *Sa*ClpP.** Effect of (*R*)- (**a**) and (*S*)-ZG197 (**b**) on the thermal stability of *Sa*ClpP and *Hs*ClpP in DSF assay, respectively. $T_m$, melting temperature. **c** Determination of (*R*)- and (*S*)-ZG197 binding to *Sa*ClpP by ITC titration. The dissociation constant ($K_d$) and stoichiometry factor (N) are indicated. Effect of activators on the thermal stability of the cellular *Sa*ClpP (**d**) and *Hs*ClpP (**e**), respectively. The *S. aureus* 8325-4 was cultured in the presence of 10 μM compound for 2 h before running CETSA (**d**), while cell lysates of HEK 293 T/17 were assayed in CETSA (**e**). The proteins in unheated samples are used as inputs and considered as 100%. Data (**a**, **b**, **d**, and **e**) are obtained from three biologically independent experiments and presented as mean ± SD (error bars). Source data are provided as a Source Data file.

since (*S*)-ZG197 fails to bind to and activate *Hs*ClpPW146A, we hypothesized that other structural factors attenuate the interaction of (*S*)-ZG197 and *Hs*ClpP mutant. Previously, it was suggested that the extra motif in the *Hs*ClpP C-terminus disturbs the ClpX chaperone's binding[48]. Particularly, we noticed the P248 and P249 residues might form a lid motif that could hinder activators' approaching and binding to the hydrophobic sites of *Hs*ClpP (Fig. 4i). This extra lid motif in *Hs*ClpP is also conserved in other eukaryotic ClpPs, like *Mus Musculus* ClpP (*Mo*ClpP) and *Danio Rerio* ClpP (*Da*ClpP). However, this C-terminal sequence is shortened and the corresponding lid motif is absent in the bacterial ClpPs, such as *Sa*ClpP and *Escherichia coli* ClpP (*Ec*ClpP) (Supplementary Fig. 2a).

We then purified the C-terminal truncated *Hs*ClpP (*Hs*ClpPΔC) and *Hs*ClpPW146A (*Hs*ClpPW146AΔC) proteins and detected our activators' effects on the two truncations. Similar to the weak effects on *Hs*ClpPW146A, (*S*)-ZG197 still minimally activates and barely stabilizes *Hs*ClpPΔC in vitro (Fig. 4j and k and Supplementary Fig. 4j). However, (*S*)-ZG197 becomes active to promote *Hs*ClpPW146AΔC mutant for α-casein hydrolysis and displays binding affinity as measured in DSF (Fig. 4j and k and Supplementary Fig. 4k), indicating the joint action of W146 and C-terminal motif in *Hs*ClpP. Interestingly, both (*R*)-ZG197 and ONC212 exhibit improved capabilities for binding to and activating on *Hs*ClpPW146AΔC compared to *Hs*ClpPW146A and *Hs*ClpPΔC (Fig. 4j and k and Supplementary Fig. 4k). Altogether, this indicates that the two distinct amino acids (I91 in *Sa*ClpP and W146 in *Hs*ClpP) in the hydrophobic pockets and the extra C-terminal lid motif in *Hs*ClpP are crucial factors that can determine the selective activation of *Sa*ClpP over *Hs*ClpP by our species-specific activators.

## Our activators exhibit antibacterial activity in a *Sa*ClpP-dependent manner

We next performed a minimum inhibitory concentration (MIC) assay in liquid culture to determine the antibacterial effects of our activators on the *S. aureus* 8325-4 strain. (*R*)-ZG197 significantly suppresses *S. aureus* with a quantified MIC of 0.5 μg/mL, which is as active as ONC212 (Fig. 5a). (*S*)-ZG197 also inhibits the growth of *S. aureus* 8325-4, and the MIC is 4 μg/mL. Both (*R*)-ZG197 and (*S*)-ZG197 exhibit much better inhibitory effects on staphylococcal growth than the hit compound ICG-001. To dissect whether our activators' antistaphylococcal activity depends on ClpP in *S. aureus*, we determined the inhibitory effects of our activators on the Δ*clpP* mutant and the complemented *clpP* or *clpPI91W* mutant strains, respectively (Fig. 5a). The selective (*R*)- and (*S*)-ZG197 activators or the global activators ICG-001 and ONC212 minimally suppress the growth of Δ*clpP* mutant strain, and all the MICs are observed to be over 256 μg/mL. The antibacterial phenotype can be rescued in the complemented *clpP* strain in the presence of ClpP activators. Unlike (*R*)-ZG197 and ICG-001 and ONC212, the antibacterial activity of (*S*)-ZG197 is significantly attenuated in the complemented *clpPI91W* mutant strain since (*S*)-ZG197 is inactive on the *Sa*ClpPI91W mutant in vitro and in cells. This indicates that our activators exhibited antibacterial activities in a *Sa*ClpP-dependent manner in *S. aureus*.

The cell division protein *Sa*FtsZ could be over-degraded upon the aberrant activation of *Sa*ClpP in the presence of activators in cells[25]. Indeed, all activators, including (*R*)-ZG197, (*S*)-ZG197, and ONC212, decrease *Sa*FtsZ abundance in the 8325-4 *S. aureus* but not in the corresponding Δ*clpP* mutant strain, as detected by the immunoblotting assay (Fig. 5b). The *Sa*FtsZ abundances are dramatically decreased

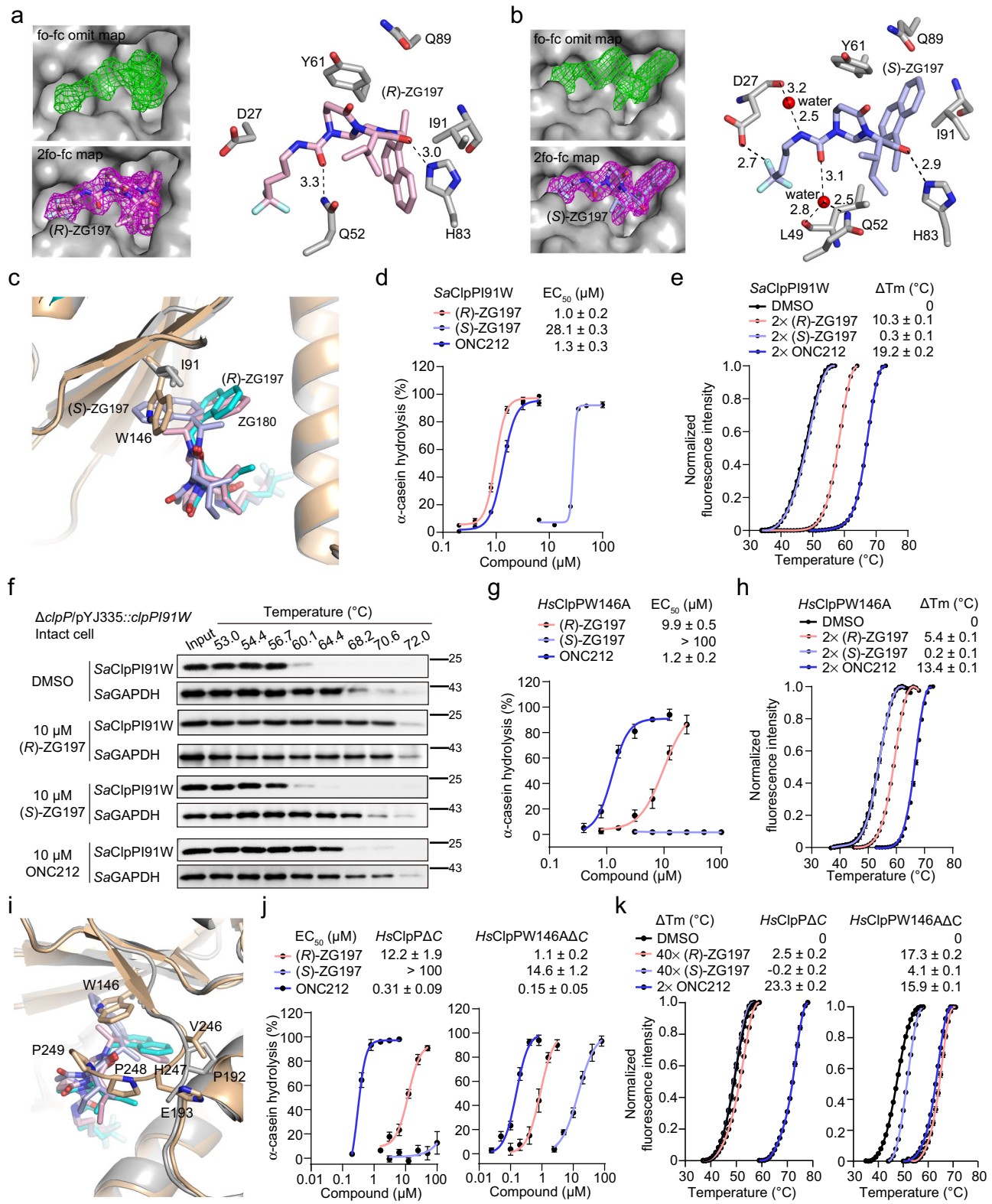

in the complemented *clpP* or *clpPI91W* mutant in the presence of the selective activator (*R*)-ZG197 or the global activator ONC212. Similar to the observation that (*S*)-ZG197 could not activate the *Sa*ClpPI91W mutant protease activity, we observed no obvious decrease in the abundance of *Sa*FtsZ in the complemented *clpPI91W* strain in the presence of (*S*)-ZG197. This indicates that our activators induced the aberrant degradation of *Sa*FtsZ in *S. aureus*, which is highly dependent on the cellular *Sa*ClpP.

The dysregulated degradation of *Sa*FtsZ could impair cell division in *S. aureus*[24,25]. We then measured the cell morphology of staphylococcal cells using a scanning electron microscope (SEM), which could reflect the inhibitory effects of our activators on staphylococcal cell division[49]. The diameters of staphylococcal cells are measured as 0.64 μm, 0.69 μm, 0.73 μm, and 0.75 μm in average for 8325-4, Δ*clpP*, and complemented *clpP* or *clpPI91W* strains in vehicle groups, respectively (Fig. 5c and Supplementary Fig. 5a). Interestingly, similar

**Fig. 4 | Identification of key sites/motif for selective *Sa*ClpP over *Hs*ClpP activation.** Close view of interactions of (*R*)- (**a**) and (*S*)-ZG197 (**b**) in the hydrophobic site of *Sa*ClpP. The fo-fc omit map for showing (*R*)- and (*S*)-ZG197 binding is contoured at 3.0 Å and colored in green, while the 2fo-fc map is contoured at 1.0 Å and colored in magenta, respectively. (*R*)- and (*S*)-ZG197 are colored in light pink and light blue, respectively. Hydrogen bonds are indicated by dark dashed lines, and the distance is labeled in Å. **c** Close view of structural alignments of (*R*)- and (*S*)-ZG197 bound to *Sa*ClpP complexes, and ZG180 bound to *Hs*ClpP in the hydrophobic pockets. ZG180, (*R*)- and (*S*)-ZG197 are colored in cyan, light pink, and light blue, respectively. *Sa*ClpP is colored in gray, while *Hs*ClpP is colored in wheat. **d** Quantitation of α-casein hydrolysis by the *Sa*ClpPI91W mutant in the presence of (*R*)-ZG197, (*S*)-ZG197, and ONC212, respectively. **e** Effect of activators on the thermal stability of the *Sa*ClpPI91W protein detected in the DSF assay. **f** Representative Western blot images showing the effects of 10 μM activators on the thermal

denaturation of the *Sa*ClpPI91W protein in intact cells of the complemented *clpPI91W* mutant strain. *Sa*ClpPI91W expression was induced in the presence of anhydrous tetracycline (ATC) at 1 ng/mL. Three biological replicates were performed for each experiment. **g** Quantitation of ClpP activators-promoted hydrolysis of α-casein by the *Hs*ClpPW146A mutant. **h** Effect of activators on the thermal stability of the *Hs*ClpPW146A mutant detected in the DSF assay. **i** Close view of the structural alignment in (**c**) shows the effect of the C-terminal residues on activators' binding and activation of *Hs*ClpP. **j** Quantitation of effect of activators-activated α-casein hydrolysis by the *Hs*ClpPΔC (left) and *Hs*ClpPW146AΔC (right) truncations, respectively. **k** Effect of activators on the thermal stability of the *Hs*ClpPΔC (left) and *Hs*ClpPW146AΔC (right) detected in the DSF assay Data (**d**, **e**, **g**, **h**, **j**, and **k**) are obtained from three biologically independent experiments and presented as mean ± SD (error bars). Source data are provided as a Source Data file.

to the inhibitory effects of FtsZ inhibitor TXA709 on staphylococcal cell division, (*R*)- and (*S*)-ZG197 treatments significantly enlarge the cell diameters of 8325-4 and the complemented *clpP* strains, with the measured diameters usually larger than 1.00 μm[50]. Shrinking, lysis, and full degradation of staphylococcal cell morphology were observed in bacteria treated with our activators. In contrast, we did not observe changes in the bacterial diameter or cell morphology when the Δ*clpP* mutant was exposed to (*R*)- and (*S*)-ZG197 activators. Treatment with (*R*)-ZG197 results in enlarged cells size in the complemented *clpPI91W* mutant strain, while (*S*)-ZG197 has no effect on bacterial diameter in this strain. We also tested the effect of our selective activators on cell divisions in Newman, a clinical isolate of methicillin-sensitive *S. aureus* strain, and in USA300, a representative MRSA isolate (Supplementary Table 2). A similar phenomenon of enlarged diameters in Newman and USA300 cells was observed in the presence of (*R*)- and (*S*)-ZG197 (Fig. 5d, Supplementary Fig. 5b, c). We demonstrated that (*R*)- and (*S*)-ZG197 exhibit suppressive effects on cell division in *S. aureus*, which is dependent on the cellular *Sa*ClpP.

Given our species-specific activators suppress the growth of *S. aureus* 8325-4, we then tested the antistaphylococcal spectrum of (*R*)- and (*S*)-ZG197 against a panel of *S. aureus* strains, including the Newman strain and multidrug-resistant MRSA strains. The USA300, NRS-1, NRS-70, NRS-100, NRS-108, and NRS-271 MRSA strains were collected from the Network on Antimicrobial Resistance in *S. aureus* (Supplementary Table 2). (*R*)-ZG197 displays strong antibacterial activity on a broad spectrum of these *S. aureus* strains, with MIC values of 0.5-2 μg/mL, while (*S*)-ZG197 suppresses the growth of these strains with moderate MIC values of 2–8 μg/mL (Fig. 5e, left panel). We also tested the antibacterial effects of our activators on five hospital-acquired, multidrug-resistant MRSA isolates in China, including XJ009, XJ036, XJ049, XJ051, and XJ052 (Supplementary Table 2). These clinical MRSA isolates have developed remarkable resistance towards several clinical antibiotics, including oxacillin, tetracycline, erythromycin, norfloxacin, clindamycin, and gentamicin, but they are all sensitive to our activators with MIC values of 0.5-1 μg/mL for (*R*)-ZG197 and 2–8 μg/mL for (*S*)-ZG197, respectively (Fig. 5e, right panel). The global activator ONC212 also exhibits low MIC values on these strains.

Similar to ONC212 and vancomycin, both (*R*)-ZG197 and (*S*)-ZG197 exhibit excellent bactericidal effects on the exponentially growing Newman strain (Supplementary Fig. 5d). We next assessed the antibacterial effects of our activators on eradicating persisters, which are cells subjected to acute stress to arouse antibiotic persistence[51]. A cohort of overnight stationary phase *S. aureus* cells that suffered from nutrient limitation are persisters and were difficult to be eradicated with conventional antibiotics[52]. Like the well-established ClpP activator ADEPs, (*R*)-ZG197, (*S*)-ZG197, rifampicin, and ciprofloxacin each alone minimally reduce the colony-forming units (CFU) of the MRSA strain of USA300 cultures (Fig. 5f and Supplementary Fig. 5e), while a combined therapy using either (*R*)-ZG197 or (*S*)-ZG197 with rifampicin or ciprofloxacin significantly reduces the number of USA300 persisters to the

limit of detection (Fig. 5f). To further investigate the suppressive effect of combined therapy on persistent *S. aureus* cells, we performed a biphasic killing assay on the USA300 strain. Similar to ADEP 4[23], we found that adding a second conventional antibiotic, such as rifampicin or ciprofloxacin, slightly impair the growth of ciprofloxacin-induced persisters, while the presence of (*R*)- and (*S*)-ZG197 *Sa*ClpP activators completely eradicate persisters to the limit of detection (Supplementary Fig. 5f). Collectively, these results verify that our activators eradicate staphylococcal persisters as efficiently as ADEP 4 when combined with conventional antibiotics.

We also screened the resistant mutants of *Sa*ClpP induced by our species-specific activators. Resistance rates are defined according to the established protocol[47]. The frequencies of spontaneous resistance induced by 4 × MIC of (*R*)-ZG197 and (*S*)-ZG197 are in the range of $10^{-7}$, which are comparable with ADEP 4 and ONC212, the global activators of ClpP. All mutation sites are randomly distributed and located outside of the hydrophobic binding pockets of ClpP (Supplementary Table 3).

## Anti-infective effects of (*R*)- and (*S*)-ZG197 in vivo

To test the anti-infective effects of our activators in vivo, we first assessed their bioactive profiles. Unlike the screening hit compound ICG-001 that was initially identified as a Wnt/β-catenin inhibitor, (*R*)- and (*S*)-ZG197 minimally disrupt the Wnt/β-catenin signaling pathway in HEK 293T/17 and HK-2 cell lines in the Wnt/β-catenin luciferase reporter assay (Supplementary Fig. 6a)[39]. We then investigated the effects of our *Sa*ClpP activators on viability of HEK 293T/17 and HK-2 cell lines in MTT assay. As anticipated, (*R*)- and (*S*)-ZG197 have minimal toxic effects on proliferation of mammalian cells, as evidenced by low inhibitory activities with high $IC_{50}$ values (Supplementary Fig. 6b). In contrast, the global activator ONC212 and our hit compound ICG-001 dramatically inhibit mammalian cell viability, especially ONC212, which has low $IC_{50}$ values of 20–60 nM. We also assessed the cytotoxic effect of our activators on zebrafish. The administration of (*R*)- or (*S*)-ZG197 at a single dose of 25 or 50 mg/kg has no cytotoxic effect on zebrafish survival (Supplementary Fig. 6c). When administrated at 100 mg/kg, ONC212-significantly cause zebrafish death, while (*R*)- or (*S*)-ZG197 is still safe on zebrafish (Supplementary Fig. 6c). This indicates the biosafety of our species-specific activators for antistaphylococcal infections in vivo.

We then evaluated the in vivo antibacterial effects of ZG197 using a zebrafish infection model, which is a widely used in vivo model for assessing antistaphylococcal agents against *S. aureus* infections[53]. We first determined an appropriate CFUs of *S. aureus* that can lead to zebrafish mortality within a reasonable time window (Supplementary Fig. 6d). We evaluated the antibacterial potency of our *Sa*ClpP activators by measuring the survival rate of zebrafish after a staphylococcal infection for five days (Supplementary Fig. 6e). A single administration of 50 mg/kg of (*R*)- and (*S*)-ZG197 significantly prolong the survival rate of USA300-infected zebrafish (Fig. 6a). A lower dose of (*R*)- or (*S*)-ZG197 at 25 mg/kg also effectively protect zebrafish against

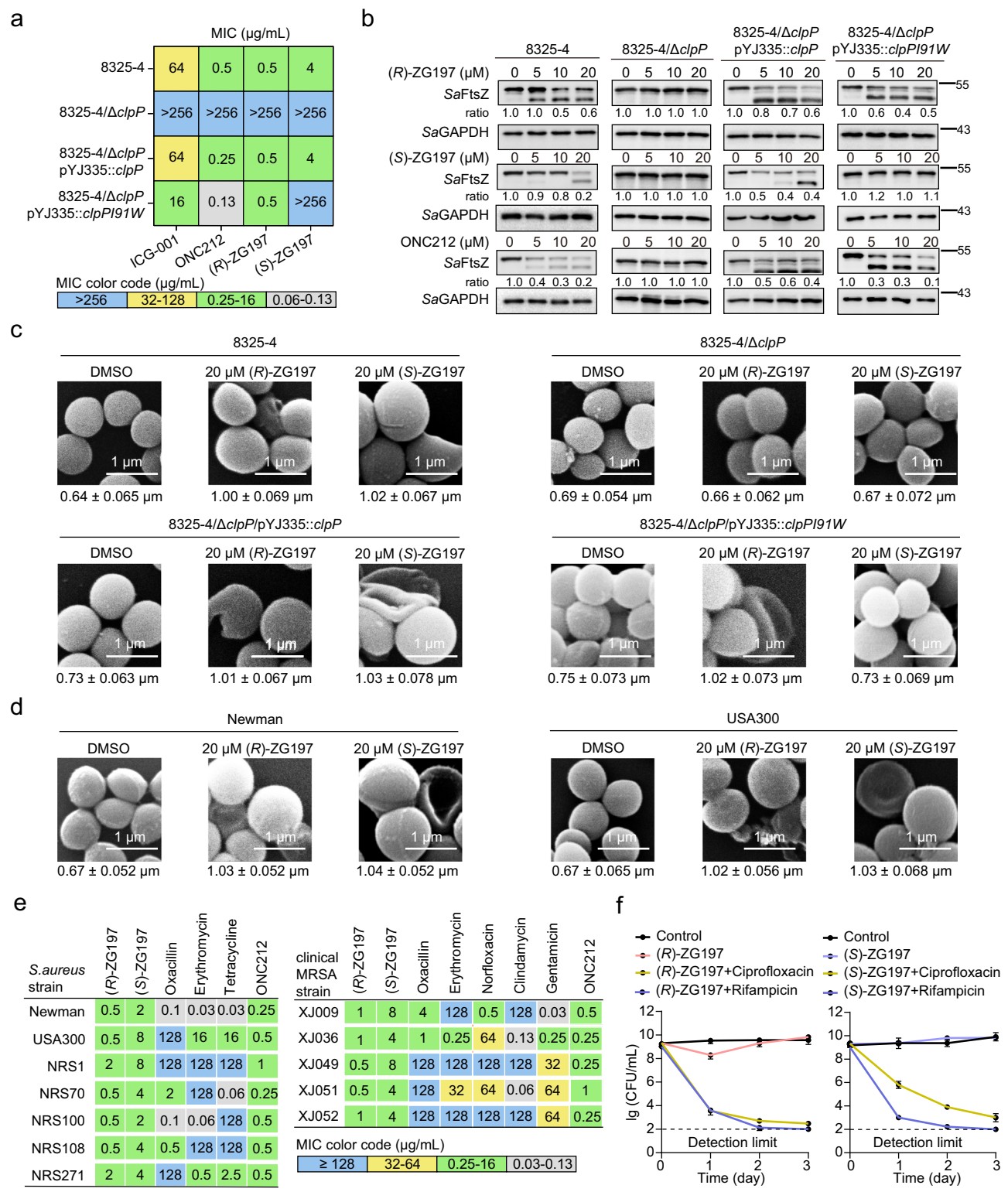

**Fig. 5 | Target dependency of (R)- and (S)-ZG197' antibacterial activity in cellular contexts. a** Assessment of ClpP activators on the growth of *S. aureus* 8325-4 with genetic backgrounds of *clpP*, Δ*clpP*, and complemented *clpP* or *I91W* mutant by MIC measurement. **b** Assessment of *Sa*ClpP activators' effect on the cellular *Sa*FtsZ abundance in the intact cells of 8325-4, Δ*clpP*, the complemented *clpP* and *clpPI91W* strains detected in Western blot. Quantitation on the band intensity of *Sa*FtsZ is normalized against the loading control of *Sa*GAPDH, and the ratio in the DMSO group is considered as 1.0. **c** Representative views of SEM micrographs showing the effects of (R)- and (S)-ZG197 on the cell morphology of *S. aureus* 8325-4 having different genetic backgrounds of *clpP*, Δ*clpP*, and complemented *clpP* or *clpPI91W*.

The scale bar represents 1 μm. **d** Representative views of SEM micrographs showing the effect of 20 μM (R)- and (S)-ZG197 on cell division of Newman and USA300 strains. The scale bar represents 1 μm. **e** Determination of MIC of the *Sa*ClpP activators against the growth of a panel of *S. aureus* strains. (R)- and (S)-ZG197, ONC212, and the clinic antibiotics oxacillin, tetracycline, erythromycin, norfloxacin, clindamycin, and gentamicin were assayed under similar conditions. The experiments were performed in triplicates. **f** Bactericidal effect of (R)- and (S)-ZG197 combined therapy on eradicating stationary phase USA300 (n = 3 biologically independent experiments). Cell numbers were recorded in CFU/mL. Data are presented as mean ± SD. Source data are provided as a Source Data file.

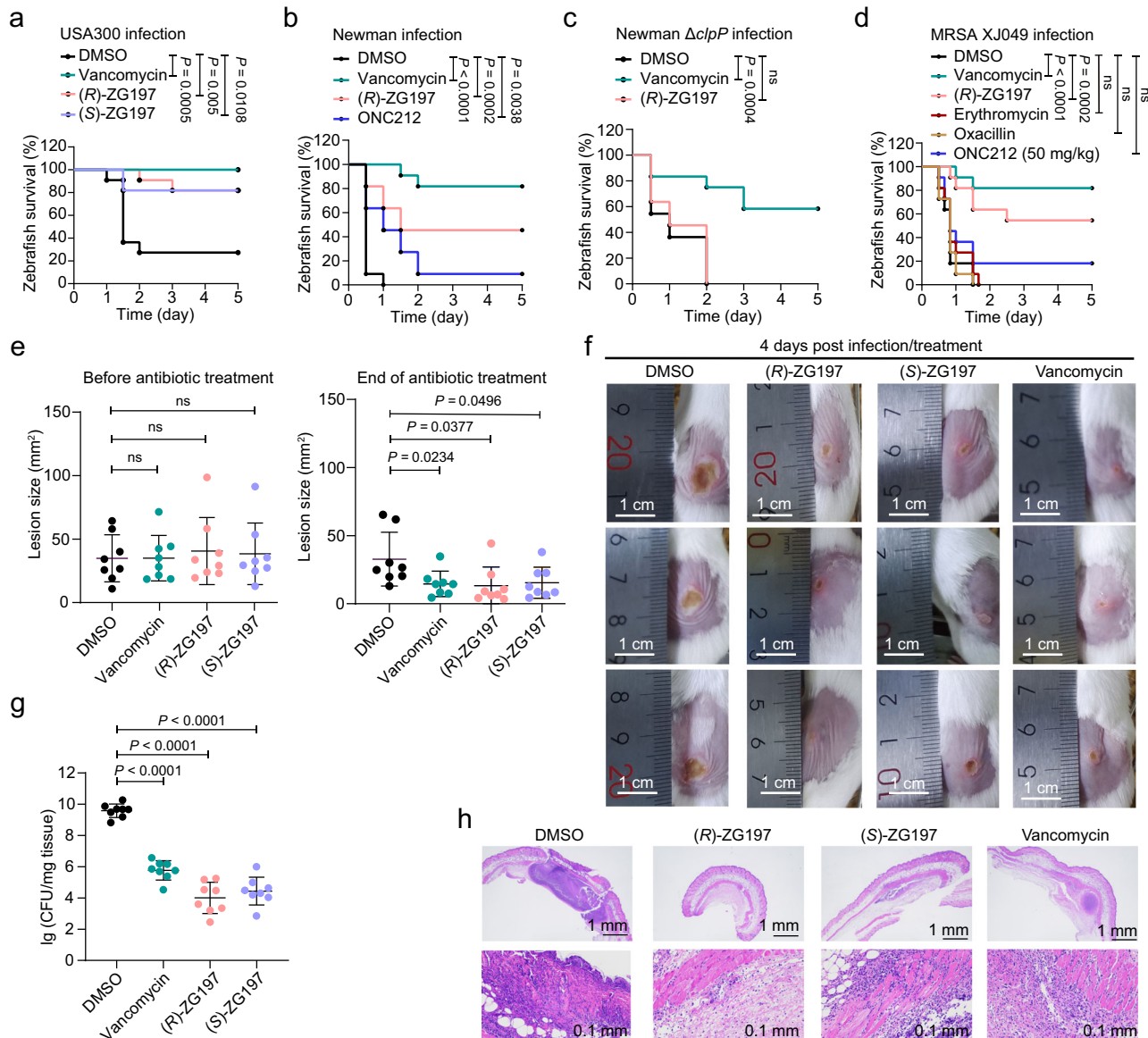

**Fig. 6 | Anti-infective effects of (*R*)- and (*S*)-ZG197 in vivo. a** Therapeutic effect of (*R*)- and (*S*)-ZG197 against USA300 infection in zebrafish. Zebrafish (*n* = 11 animals) were infected with $7 \times 10^6$ CFU of USA300, and compounds were administered at a single dose of 50 mg/kg. DMSO (5%) and vancomycin were administered as controls. Therapeutic effect of (*R*)-ZG197 against lethal infection by Newman (**b**) and the Δ*clpP* mutant (**c**). Zebrafish (*n* = 11 animals) were infected with $7 \times 10^7$ CFU of Newman (**b**) or $3 \times 10^7$ CFU of the Δ*clpP* mutant (**c**), and compounds were administered to infected zebrafish at 50 mg/kg. Vancomycin and ONC212 were used as controls. **d** Therapeutic effect of (*R*)-ZG197 on the clinical MRSA isolates infected zebrafish. Zebrafish (*n* = 11 animals) were infected with $5 \times 10^7$ CFU of XJ049, and compounds were administered at a single dose of 50 mg/kg. DMSO (5%), antibiotics (vancomycin, erythromycin, and oxacillin), and the global activator ONC212 were used as controls. **e** Quantitation of necrotic skin lesion size of initial (left panel) and 4 days post-infection (right panel). Vancomycin was used as a positive control. Data are shown as mean ± SD (*n* = 9 animals). **f** Representative images showing necrotic skin lesions 4 days post-infection. The scale bar represents 1 cm. **g** Effects of our activators on bacterial counts in the skin samples (*n* = 8 biologically independent samples). Skin tissue was excised from mice infected with USA300 4 days post-infection and plated onto TSA to enumerate CFUs. **h** Micrographs of H&E-stained skin tissues after the indicated treatments. The scale bar represents 1 mm (Top) and 0.1 mm (Bottom). Statistical differences were analyzed using Log-rank tests (**a**–**d**) and two-tailed unpaired Student's *t*-tests (**e**, **g**). Exact *P* values are provided. ns, no significance. Data (**e**, **g**) are presented as mean ± SD (error bars). Source data are provided as a Source Data file.

USA300-induced lethal infection (Supplementary Fig. 6f). Similarly, the administration of 25 mg/kg or 50 mg/kg (*R*)-ZG197 effectively improve the survival rate of Newman-infected zebrafish, which has a better survival rate than ONC212 (Supplementary Fig. 6g and Fig. 6b). However, (*R*)-ZG197 loses therapeutic effects on zebrafish infected with the Δ*clpP* mutant strain, while vancomycin is still active against this infection (Fig. 6c). This indicates that the promising anti-staphylococcal effect of (*R*)-ZG197 on infected zebrafish is dependent on *Sa*ClpP in vivo. A single administration of 50 mg/kg of (*R*)-ZG197 produce a significant therapeutic effect on zebrafish infected with the clinical MRSA strain XJ049. In contrast, this strain is resistant to the

treatment of clinically used antibiotics oxacillin and erythromycin (Fig. 6d). It is unsurprising that the global activator ONC212 at 50 mg/kg fail to prolong the survival rate of zebrafish infected with MRSA XJ049 likely due to its cytotoxic side-effects by targeting *Da*ClpP (Fig. 6d).

Lastly, we evaluated the in vivo anti-infective efficacy of our activators in murine skin *S. aureus* infection models[54]. Mice skin was inoculated with $2.5 \times 10^6$ CFUs of USA300 and a lesion appeared after 16 h. The initial lesion areas in each group are comparable (Fig. 6e). Subcutaneous injection of (*R*)- and (*S*)-ZG197 at 7.5 mg/kg twice a day causes a smaller necrotic lesion size in mice compared with the vehicle

control (Figs. 6e, f). Moreover, we observed a significant reduction in bacterial load when we excised the lesions of mice treated with (*R*)- and (*S*)-ZG197 compared to the vehicle control (Fig. 6g). Histological analysis using hematoxylin and eosin (H&E) staining also exhibits a decrease in the necrotic area and inflammatory infiltrates in the skin tissues when treated with our activators or vancomycin, while the vehicle-treated sample displays dense inflammatory infiltrates (Fig. 6h). Collectively, we identify our activators as promising antibiotics against multidrug-resistant MRSA infections in vivo but with minimal cytotoxic effects.

## Discussion

Aberrant activation of bacterial ClpP protease is a promising method of treating nosocomial infections with antibiotic-resistant Gram-positive bacteria. The desired *Sa*ClpP activator should preferably target the staphylococcal ClpP rather than disturb the proper function of the host ClpP. However, earlier work identified several global activators for the ClpP proteases of multiple species, including ADEP analogs and ONC212, which would inevitably produce unintended off-target effects and hamper the application in drug discovery for anti-MRSA infection. Therefore, identifying distinct chemical scaffolds that selectively activate *Sa*ClpP will highlight its anti-infective therapeutic potential for MRSA infections using a chemoactivation strategy.

We use HTS for compounds that promote the ClpP protease activity and experimental validation to identify *N*-benzyl- and naphthalen-1-ylmethyl-disubstituted peptidomimetic compound (ICG-001) as an activator with a moderate activity for both *Sa*ClpP and *Hs*ClpP. Synthetic optimization of ICG-001 analogues led to discovery of (6*S*,9a*S*)-6-((*S*)-sec-butyl)-8-(naphthalen-1-ylmethyl)-4,7-dioxo-*N*-(4,4,4-trifluorobutyl)hexahydro-2*H*-pyrazino[1,2-a]pyrimidine-1(6*H*)-carboxamide (ZG180) with significantly improved activities on both proteases. The incorporation of methyl substituent group on ZG180 makes (*R*)- and (*S*)-ZG197 the most potent and highly selective activators for *Sa*ClpP over *Hs*ClpP. Our activators increase the aberrant degradation of *Sa*FtsZ protein in cells, inhibit staphylococcal cells division, and display significant antibiotic efficacy in preventing the lethal outcome of *S. aureus* bacteremia in vivo. Earlier work demonstrated ICG-001 as an inhibitor for the Wnt/β-catenin signaling pathway and exerted antiproliferative effects in xenografted murine models. Nevertheless, our activators minimally inhibit the viability of mammalian cell lines, which should be explored further during the clinical development of antibiotics.

Our identification of (*R*)- and (*S*)-ZG197 as species-specific activators for *Sa*ClpP provides structural insights into the development of selective binding and specific chemoactivation of certain ClpP. The bulky W146 of *Hs*ClpP affords an unfavorable steric clash with the hindrance naphthyl substituent in (*S*)-ZG197 activator, therefore prevents ligands' binding to *Hs*ClpP. This indicates W146/I91 residue is a critical factor that partially determines the species-selectivity of our activators. In addition to the observation that W146 in *Hs*ClpP represents a critical point for activator discrimination, our work also reveals W146's joint action with the C-terminal motif in *Hs*ClpP, as evidenced by the recovered activities of (*S*)-ZG197 and the improved activities of (*R*)-ZG197 towards *Hs*ClpPW146AΔC's binding and activation compared to either the *Hs*ClpPW146A mutant or the *Hs*ClpPΔC truncation. Along with previously published research, this shows that the key structural factors, including spatial difference in hydrophobic pockets and coordination of other amino acids, significantly contribute to the discrimination of ClpP activators. Additionally, variant activators respond differently to the bacterial and mitochondrial ClpP proteins.

Chemoactivation of bacterial ClpP efficiently switches it into an uncontrollable state to trigger excessive degradation of numerous substrates that are crucial to bacteria and ultimately destroy the pathogenic bacteria. We demonstrate that both (*R*)-ZG197 and (*S*)-ZG197 selectively bind to *Sa*ClpP over *Hs*ClpP, induce the activator-

bound *Sa*ClpP in the fully extended, active conformation for non-selective hydrolysis of essential proteins, such as FtsZ. Although bacterial ClpP is not essential for survival, the ClpP activators are inclined to induce ClpP mutation and thus develop resistance. Therefore, combination therapy is likely the best method of therapeutic application for *Sa*ClpP activators, which could lower target-based resistance, broaden the antibiotic spectrum, and reduce dosage and unwanted side-effects. We also show that either (*R*)-ZG197 or (*S*)-ZG197 used in combination with rifampin or ciprofloxacin dramatically eradicates the population of persisters to the detection level.

Given highly conserved ClpP proteases among multiple species, it must be investigated whether our activators can similarly switch other ClpP proteases from different species into gain-of-function states, especially with Gram-negative bacteria and gut microbiota. Further synthetic optimizations are needed to improve the antimicrobic capabilities of our species-specific *Sa*ClpP activators to treat MRSA infections. Assessment of the anti-infective effects of our activators on additional in vivo animal infectious models would fully characterize the antibacterial potency of these activators.

In conclusion, we report (*R*)- and (*S*)-ZG197 as selective *Sa*ClpP activators and identify the critical residues/motif for selectively promoting *Sa*ClpP rather than *Hs*ClpP. Our data provide mechanistic insights into the chemoactivation of ClpP proteases and accelerate the discovery of species-specific antibiotic drugs that can selectively disturb the function of staphylococcal protease. This work has the potential to inspire promising strategies for treating MRSA infections and contributes to the generation of potent synthetic antibiotic scaffolds with minimal cytotoxicity to the host.

## Methods

### Animals

Wild-type zebrafish (7 to 11 months old, $300 \pm 50$ mg, regardless of gender) were purchased from the Xiaoguan aquarium (Shanghai, China) and maintained in a 10-L tank at ambient temperature with regular feeding. Similar size and random sex distribution of zebrafish were used for *S. aureus* infection. Female BALB/c mice (6 to 8 weeks old, 18–20 g) were purchased from Zhejiang Vital River Laboratory Animal Technology Co., Ltd. All mice were housed on an *ad libitum* diet in the specific pathogen-free facility at Shanghai Public Health Clinical Center. Housing conditions of dark/light cycle 12 h, ambient temperature 20–26 °C, humidity 40–60% were applied on mice.

### Antibodies

Antibodies of *Sa*ClpP (1:5,000, Cat#C11185), *Sa*FtsZ (1:5,000, Cat#C11186), and *Sa*GAPDH (1:5,000, Cat#C1399) were generated by Shanghai Immune Biotech Ltd using the purified protein as the antigen and validated by ELISA experiments. Antibodies of *Hs*ClpP (1:2,000, Clo#EPR7133, Cat#ab124822, Lot#GR3210822-8, Abcam), β-actin (1:5,000, Clo#2D4H5, Cat#66009-1-Ig, Lot#10004156, Proteintech), HRP-conjugated goat anti-rabbit IgG (1:10,000, Cat#CW0103, Cwbio), and HRP-conjugated goat anti-mouse IgG (1:10,000, Cat#CW0102, Cwbio) were commercially purchased.

### Bacterial strains and mediums

The bacterial strains used in this study are provided in Supplementary Table 2. Mediums of Tryptic Soy Broth (TSB, OXOID), Tryptone Soya Agar (TSA, OXOID), Brian Heart Infusion (BHI, OXOID), Mueller-Hinton agar (MHA, Dalian Meilun Biotech) and Mueller-Hinton Broth (MHB, Dalian Meilun Biotech) mediums were used for cultivation of *S. aureus*. Luria Bertani (LB) in broth or LB agar mediums were used for *Escherichia coli* (*E. coli*) growth.

### Chemicals

ZG180, (*R*)- and (*S*)-ZG197 were synthesized in lab and fully characterized. ONC212 and ADEP 4 were purchased from Topscience and

ChemPartner (Shanghai, China), respectively. Antibiotics of vancomycin, spectinomycin, erythromycin, ciprofloxacin, rifampicin, tetracycline, norfloxacin, clindamycin, and gentamicin, were purchased from Dalian Meilun Biotech. Oxacillin was purchased from Sigma-Aldrich. Anhydrous tetracycline (ATC) was purchased from APExBIO.

## Mammalian cell lines and cultures

The HEK 293T/17 (CRL-11268) and HK-2 (CRL-2190) cell lines were purchased from the American Type Culture Collection (ATCC). Cells were cultured in Dulbecco's Modified Eagle Medium (DMEM, Corning) supplemented with 10% fetal bovine serum (FBS, GIBCO) and 1% penicillin-streptomycin (Corning). Cells were cultured at 37 °C in a humidified, 5% $CO_2$-containing atmosphere incubator (Thermo Scientific). All cells have been validated by short tandem repeat (STR) profiling and were regularly checked to be mycoplasma-free.

## Cloning and protein purification

The plasmid pET28b-SaClpP was used for SaClpP expression[49]. E. coli BL21 (DE3) Gold strains were transformed with the plasmid in the presence of 30 μg/mL kanamycin. When $OD_{600}$ (absorbance at the wavelength of 600 nm) reaches 0.6-0.8, protein expression was induced by 0.5 mM Isopropyl β-D-1-thiogalactopyranoside (IPTG) at 30 °C for 4 h. Then cell pellets were harvested and stored at −80 °C. Cells were lysed and the supernatant was subjected to 5 mL HisTrap™ HP (GE Healthcare) column in binding buffer (50 mM Tris-HCl, pH 8.0, 100 mM NaCl, and 50 mM imidazole), and then eluted with elution buffer (50 mM Tris-HCl, pH 8.0, 100 mM NaCl, and 400 mM imidazole). The eluted proteins were concentrated with Amicon Ultra Centrifugal Filters with 10 kDa cutoff (Merck-Millipore) and subjected to gel filtration on an AKTA purifier system with a HiLoad 16/60 Superdex 200 column (GE Healthcare) in 20 mM HEPES, pH 7.0 and 100 mM NaCl. The site-directed mutation of SaClpPI91W was established according to the Quikchang Site-Directed mutagenesis Kit (Stratagene). A similar procedure was applied on SaClpPI91W purification.

The plasmid pET28a-SaFtsZ was used for SaFtsZ expression[49], and transformed into E. coli BL21 (DE3) Gold in the presence of 30 μg/mL kanamycin. When $OD_{600}$ reaches 0.6, expression of SaFtsZ was induced by 0.5 mM IPTG for 12 h at 16 °C. Then cell pellets were lysed, and the supernatant was bound with 5 mL HisTrap™ HP column in binding buffer (50 mM Tris-HCl, pH 8.0, 200 mM NaCl, and 50 mM imidazole), and then eluted with elution buffer (50 mM Tris-HCl, pH 8.0, 200 mM NaCl, and 400 mM imidazole). The eluted proteins were concentrated with Amicon Ultra Centrifugal Filters with 10 kDa cutoff (Merck-Millipore) and was stored at −80 °C.

The pPSUMO-CLPP$_{ΔN56}$ plasmid was transformed to E. coli Rosetta (DE3) cells for HsClpP expression[40]. The cells were cultured at 37 °C until $OD_{600}$ reached 0.6-0.8 and the expression of HsClpP was induced by 0.5 mM IPTG at 37 °C for 4 h. Cells were lysed in binding buffer (20 mM Tris-HCl, pH 7.5, 500 mM NaCl, 10 mM imidazole, and 10% glycerol). The supernatant was then passed through a 5 mL HisTrap™ HP column. After washing with binding buffer, target protein was eluted with elution buffer (20 mM Tris-HCl, pH 7.5, 500 mM NaCl, 500 mM imidazole, and 10% glycerol). The collected protein was then concentrated and resuspended in binding buffer. Then, the ULP1 protease was added to cleave the N-terminal His-SUMO tag at 4 °C overnight with gentle shaking. The remaining fractions were then passed through HisTrap™ HP column for removing the His-SUMO tag and the His-tagged ULP1. Further purification was conducted through a HiLoad 16/60 Superdex 200 column in 20 mM Tris-HCl, pH 7.5, 500 mM NaCl, and 5% glycerol. The construction of plasmids for HsClpPW146A, HsClpPΔC, and HsClpPW146AΔC were performed according to the Quikchang Site-Directed mutagenesis Kit. The purifications of HsClpPW146A, HsClpPΔC, and HsClpPW146AΔC were similar to that of HsClpP. The proteins were stocked at −80 °C in 30% glycerol.

The primer's sequences used for constructing plasmids for mutants are shown as follows:

*SaclpPI91W-F:* GATGTTCAAACATGGTGTATCGGTATGGC
*SaclpPI91W-R:* GCCATACCGATACACCATGTTTGAACATC
*HsCLPPW146A-F:* CCGATTTGTACCGCCTGTGTGGGTC
*HsCLPPW146A-R:* GACCCACACAGGCGGTACAAATCGG
*HsCLPPΔC-F:* GTGCTGGTTCATTAGCCGCAGGATGGTGAAGATG
*HsCLPPΔC-R:* CATCTTCACCATCCTGCGGCTAATGAACCAGCAC

## SDS-PAGE based α-casein hydrolysis assay

Ten microgram SaClpP, SaClpPI91W, HsClpP or HsClpPW146A were dissolved in 50 μL buffer containing 25 mM HEPES-KOH (pH 7.6), 200 mM KCl, 5 mM MgCl₂, 1 mM EDTA, 10% (v/v) glycerol, and 2 mM DTT, while 10 μg HsClpPΔC or HsClpPW146AΔC was dissolved in 50 μL buffer containing 25 mM HEPES-KOH (pH 7.6), 5 mM MgCl₂, 5 mM KCl, 0.03% Tween-20, 10% glycerol, and 2 mM DTT. Various concentrations of compound (final concentration of DMSO at 1%) were preincubated at room temperature for 15 min. Then 0.72 mg/mL α-casein was added, and the reaction was performed at 37 °C for 2 h. The samples were mixed with SDS loading buffer and analyzed by 12% SDS-PAGE followed by Coomassie Brilliant Blue R-250 staining. The $EC_{50}$ values were calculated by GraphPad Prism 8 from three biological replicates.

## High-throughput screening (HTS) for SaClpP activators

The tetradecameric SaClpP protein (0.6 μM) in PD buffer (25 mM HEPES-KOH (pH 7.6), 200 mM KCl, 5 mM MgCl₂, 1 mM EDTA, 10% (v/v) glycerol, and 2 mM DTT) and compounds (40 μM) were added into the black flat-bottomed 384-well plates. After incubating at 37 °C for 30 min, fluorescence-labeled substrate FITC-casein (0.048 mg/mL) (Sigma) was added, and the hydrolysis reaction was initiated at 37 °C. After 2 h, the fluorescence signals were recorded at the wavelength of 485 nm for excitation and 535 nm for emission using a microplate reader (TECAN).

## Synthesis and characterization of ZG180 and (R)- and (S)-ZG197

ZG180 and (R)- and (S)-ZG197 were synthesized in this lab and the synthetic methods along with the characterizations are provided in Supplementary methods and Supplementary Fig. 7–23.

## Crystallization and structure determination

The SaClpP and HsClpP proteins were crystalized by the sitting drop vapor diffusion method at 16 °C. The sample of 10 mg/mL protein was mixed with 5-fold concentration of the compound of ZG180, (R)- or (S)-ZG197 (contain 5% DMSO) and incubated for 30 min at 4 °C. The drops contained 1 or 2 μL SaClpP sample and 1 μL reservoir solution of 30%(v/v) 2-methyl-2,4-pentanediol, 0.1 M sodium acetate/HCl (pH 8.0), 20 mM calcium chloride in the presence of ZG180, or 1 μL reservoir solution of 30% v/v (+/−)-2-methyl-2,4-pentanediol, 0.1 M sodium acetate trihydrate (pH 4.6), 200 mM sodium chloride in the presence of (R)- and (S)-ZG197. The HsClpP sample at 10 mg/mL was incubated in the presence of 5-fold concentration of ZG180 on ice for 30 min and mixed with a reservoir solution of 20% w/v polyethylene glycol 3350, 0.2 M sodium malonate (pH 5.0). These crystals grew up after one week to more than one month. Then the crystals were fished and soaked in 20% (v/v) glycerol for 30 s following with storing in liquid nitrogen.

Diffraction data of ZG180/SaClpP and (R)-ZG197/SaClpP were collected via Bluice at Shanghai Synchrotron Radiation Facility (SSRF) beamline BL19U1[55]. X-ray data were processed via HKL2000 program suite. Diffraction data of ZG180/HsClpP or (S)-ZG197/SaClpP were collected via Finback at SSRF beamline BL02U1 and automatically processed by Aquarium[56]. The structures were resolved using the SaClpP (PDB code 3STA) and HsClpP (PDB code 1TG6) as the search model and models were built in COOT[43,48,57]. Finally, the data was refined with the program REFMAC5[58]. The free reflections were automatically selected and applied in all refinements. Structure alignments were performed in PyMOL[59].

## In vitro SaFtsZ degradation assay

The SaFtsZ degradation was analyzed in SDS-PAGE in vitro. Briefly, 10 μg SaClpP was dissolved in 50 μL PD buffer. After adding various concentrations of compound (final concentration of DMSO at 1%), the mixture was incubated at room temperature for 15 min. Then SaFtsZ (1.5 μM) was added, and the reaction was performed at 37 °C for 2 h. The samples were mixed with SDS loading buffer and analyzed by 12% SDS-PAGE followed by Coomassie Brilliant Blue R-250 staining. The $EC_{50}$ values were calculated by GraphPad Prism 8 from three biological replicates.

The cell lysate of 8325-4/ΔclpP strain was also used in SaFtsZ degradation assay. An overnight staphylococcal culture of 8325-4/ΔclpP was diluted 1:500 and cultured to early-mid exponential phase ($OD_{600}$ = 0.4–0.6) at 37 °C. Culture was then washed with PD buffer for three times followed by cell lysis using 0.75 U Lysostaphin (Sigma). Then the cell lysate was centrifuged at 12,396 × g and the supernatant was collected and divided into 50 μL. Nine micromolar bovine serum albumin (BSA, Dalian Meilun Biotech), SaClpP or HsClpP protein and various concentrations of compound was added into the system. After an incubation at 37 °C for 2 h, samples were mixed with SDS loading buffer and heated at 100 °C for 10 min. Then the samples were loaded to the SDS-PAGE for detecting cellular SaFtsZ abundance. The proteins were transferred to nitrocellulose membrane (Millipore) and blocked with TBST containing 5% skim milk at room temperature for 1 h, followed by incubation with the primary antibodies of SaFtsZ (1:5,000) and SaGAPDH (1:5,000) at 4 °C overnight. After washing with TBST, HRP-labeled goat anti-rabbit IgG (1:10,000) was added for Western Blot analysis.

## Differential scanning fluorimetry (DSF)

The SaClpP or SaClpPI91W protein was dissolved in 100 mM HEPES, pH 7.0, 100 mM NaCl to a final concentration of 2 μM (monomeric concentration), while HsClpP, HsClpPΔC, HsClpPW146A or HsClpPW146AΔC was dissolved into the assay buffer containing 20 mM Tris-HCl, pH 7.5, 500 mM NaCl, and 5% glycerol to a final concentration of 2 μM (monomeric concentration). Compounds in varying concentrations (containing 1% DMSO) were added into the assay buffer. After incubation at room temperature for 10 min, 5 × SYPRO Orange dye (Invitrogen) was added into each sample. The detection was recorded in a CFX96 Real-Time System (Bio-Rad) at the wavelength of 492 nm for excitation and 610 nm for emission. The samples were heated from 25 °C to 95 °C at a rate of 1 °C/min. Data analysis was performed with Bio-Rad CFX Manager software and GraphPad Prism 8. The $T_m$ of proteins in compound-treated groups were referenced against that in DMSO group.

## Isothermal titration calorimetry (ITC)

ITC experiments were performed in a buffer containing 20 mM HEPES, 100 mM NaCl, pH 7.0, 5% DMSO at 25 °C and with constant stirring at 750 rpm on a MicroCal iTC200 system (GE Healthcare) according to the reported method[44,60]. The proteins and compounds were dissolved in the exact same buffer. The cell was filled with 50 μM (R)- or (S)-ZG197, and the syringe was filled with SaClpPI91W at 500 μM or SaClpP at 700 μM. After an automatic equilibration of system, 2 μL solution in syringe was injected into the cell to trigger binding reaction and producing the characteristic peak sequence in the recorded signal. Data analysis including baseline correction and evaluation was carried out with OriginPro 8.5 ITC. Fits were carried out considering all injections except the first injection to calculate the maximum heat and the value of equilibrium dissociation constant ($K_d$).

## Biolayer interferometry (BLI) assay

This experiment was performed using Octet RED96 (ForteBio). The SaClpP and HsClpP proteins were biotinylated in PBS buffer at room temperature for 1 h. Then the biotinylated proteins were collected using a PD MiniTrap™ G-25 Desalting Column (cytiva). The streptavidin biosensors were incubated in PBST (PBS with 0.5% Tween-20) containing 0.5% DMSO for 10 min followed by loading with 50 μg/mL biotinylated SaClpP or HsClpP. The reference control applied a duplicate set of sensors that was incubated only in PBST with 0.5% DMSO. Various concentrations of (R)- and (S)-ZG197 were used to calculate the binding curve. The detection was conducted with a standard protocol in 96-well black plates with a total volume of 200 μL at 30 °C. BLI data were collected using Octet Acquisition 11.0. The signals were analyzed by a double reference subtraction protocol to calculate the binding kinetics using Octet Analysis 11.0. The $K_d$ values were calculated from the steady-static fit curve.

## Plasmid construction and protein overexpression in S. aureus

To construct the plasmid for the constitutive expression of SaClpP in S. aureus, a DNA fragment, covering the clpP gene and its upstream 29 base pairs, was amplified from S. aureus 8325-4 genomic DNA and cloned into pYJ335. The SaclpPI91W mutant was constructed according to QuikChange II site-directed mutagenesis kit with primers SaclpPI91W-F and SaclpPI91W-R. For plasmid maintenance, 100 μg/mL ampicillin was used in E. coli, and 10 μg/mL erythromycin was used in S. aureus. The plasmid was extracted from DH5α and subsequently transformed into electrocompetent S. aureus RN4220, from which the plasmid was extracted again and finally transformed into electrocompetent S. aureus 8325-4/ΔclpP cells. Erythromycin was used at 10 μg/mL for plasmid selection, and 1 ng/mL ATC was added to induce protein expression.

SaclpP-F1 primer: GTAACAGTTATTACAAGGAGG
SaclpP-R1 primer: AGGGGGGCCCTTATTTTGTTTCAGG
SaclpPI91W-F primer: GATGTTCAAACATGGTGTATCGGTATGGC
SaclpPI91W-R primer: GCCATACCGATACACCATGTTTGAACATC

The protein overexpression was detected using Western blot. An overnight culture of bacteria was diluted at 1:1000 in TSB with aeration at 220 rpm. When $OD_{600}$ reached 0.6, 1 mL culture was collected and centrifuged at 12,396 × g for 3 min. The sediments were washed with PBS for three times and resuspended in 100 μL PBS with 0.75 U Lysostaphin (Sigma) at 37 °C for 30 min. The samples were then centrifuged at 12,396 × g, 4 °C for 30 min and the supernatants were collected. Protein concentration was quantified with BCA Protein Assay Kit (Beyotime). The samples were diluted with SDS loading buffer, heated at 100 °C for 10 min, and loaded to the SDS-PAGE for analysis. The proteins were transferred to nitrocellulose membrane (Millipore) and blocked with TBST containing 5% skim milk at room temperature for 1 h, followed by incubation with the primary antibodies of SaClpP (1:5,000) and SaGAPDH (1:5,000) at 4 °C overnight. HRP-labeled goat anti-rabbit IgG (1:10,000) was then added. The enhanced chemiluminescence (ECL) was used to autoradiography, and the images were captured with GE ImageQuantLAS4000.

## SaFtsZ degradation in intact cells

S. aureus 8325-4 strains with genetic backgrounds of clpP, ΔclpP, and the complemented clpP or I91W mutant were assayed for SaFtsZ degradation in intact cells. An overnight culture was diluted at 1:200 and incubated with varying concentrations of each compound (0, 5, 10, and 20 μM, final concentration of DMSO at 1%). When $OD_{600}$ reaches 0.7, 300 μg/mL spectinomycin was added and the reaction was conducted at 37 °C, 220 rpm for 3 h. Then 1 mL of each sample was taken and washed with PBS three times. The cells were resuspended and lysed with 0.75 U Lysostaphin at 37 °C for 30 min. The supernatant of lysed cell extracts was separated, and proteins' concentration were determined using BCA Protein Assay Kit followed by Western blot analysis. Quantitative analysis was performed using ImageJ software. Degree of SaFtsZ degradation was determined by measuring the band intensity of SaFtsZ against SaGAPDH and the DMSO panel was considered as 1.0.

## Screening for the resistant mutants

Single colonies of *S. aureus* 8325-4 were picked up and grown at TSA with shaking overnight. The culture was diluted with 1:1000 and grown to $OD_{600} = 0.6$. Then the culture was resuspended in PBS and $5 \times 10^6$ cells were spread on MHA plates containing (*R*)-ZG197, (*S*)-ZG197, ADEP 4 or ONC212 at $4 \times MIC$ of each compound. After 24 h, resistant colonies were randomly picked for MIC determinations and sequencing. The *SaclpP* gene was amplified by PCR and sequenced for determining the mutant site.

The sequences of the primers used for amplifying *SaclpP* are shown as follows:

*SaclpP*-F2 primer: GTTATTACAAGGAGGAAAT
*SaclpP*-R2 primer: CAGACAGCTTAGTCTACTC

## Scanning electron microscopy (SEM)

Overnight cultures of *S. aureus* strains were diluted at 1:200 in fresh TSB medium and incubated with DMSO or 20 μM (*R*)- and (*S*)-ZG197 then shaken at 37 °C, 220 rpm for 3 h. The cultures were washed by PBS three times and resuspended to $OD_{600}$ of 1.5. Then 50 μL of cultures were taken and fixed on 13 mm cover glass with 300 μL 2.5% glutaraldehyde (TED PELLA) in 24 well plate (Corning) at 4 °C overnight. On the next day, the glutaraldehyde in each well was removed and washed with PBS three times. Then the samples were dyed with osmic acid carefully for 30 min followed by washing with $ddH_2O$ three times. Samples were dehydrated in a graded series of ethanol (30, 50, 70, 80, 95, 100%) for 10 min with gentle shaking for each step. Samples were then subjected to critical point drying with liquid $CO_2$ (Leica EM) to maintain original appearance of samples. Dried samples were coated with a gold film by sputter coating (Leica EM). Samples were imaged in a FEI Quanta 250 at an acceleration voltage of 15 kV. Images were recorded and contrast and brightness were adjusted with Adobe Photoshop CS5.

## Cellular thermal shift assay (CETSA)

Overnight cultures of *S. aureus* 8325-4 or the rescued *SaclpPI91W* strains were diluted at 1:200 in fresh TSB and were incubated with DMSO or 10 μM compounds, including (*R*)-ZG197, (*S*)-ZG197, and ONC212, for 2 h. One milliliter of each culture was collected and washed with PBS for three times. Then the samples were resuspended in PBS containing cocktail inhibitors (Sangon Biotech) and divided into 50 μL aliquots and heated at different temperatures for 3 min using a thermal cycler (BIO-RAD). The bacteria were then lysed with 0.75 U Lysostaphin at 37 °C for 30 min followed by three freeze-thaw cycles with liquid nitrogen to crack bacteria completely. The soluble lysates were collected by centrifugation at $12,396 \times g$ for 20 min at 4 °C and the supernatants were analyzed by Western blot. Primary antibodies of *Sa*ClpP (1:5,000) and *Sa*GAPDH (1:5,000) were used. Three independent experiments were performed, and one of the representative results is shown.

HEK 293 T/17 cells were digested by trypsin and collected. The cell sediments were resuspended in PBS containing cocktail inhibitors (Sangon Biotech) and lysed by three freeze-thaw cycles with liquid nitrogen. The lysed samples were then centrifuged at $12,396 \times g$ for 30 min to remove the debris. After centrifugation, the supernatant was incubated with 10 μM compounds, including (*R*)-ZG197, (*S*)-ZG197, and ONC212, for 20 min and then transferred into the 0.2 mL PCR tubes followed by denaturing at indicated temperatures for 3 min. Then the soluble supernatants were collected by centrifugation and were analyzed by Western blot. Primary antibodies of *Hs*ClpP (1:2,000) and β-actin (1:5,000) were used. Three independent experiments were performed, and one of the representative results is shown.

## Minimum inhibitory concentration (MIC) assay

Cultures were grown in MHB medium at 37 °C with shaking at 220 rpm from overnight until $OD_{600}$ reached 0.4-0.6, and then diluted into MHB medium to give a final $OD_{600}$ of 0.025. Compounds at various concentrations were added to diluted bacterial cultures (100 μL/well final volume and final DMSO concentration of 1%) in triplicates. The lowest concentration of an antibiotic that inhibited the visible growth of bacteria was recorded after incubation at 37 °C for 18 h. MIC values were determined by three independent experiments.

## Time killing assay

An overnight culture of USA300 was diluted in TSB with a final $OD_{600}$ of 0.025 and cultivated to mid-logarithmic phase ($OD_{600}$ of 0.6) at 37 °C, 220 rpm. Subsequently, cells were diluted to $1 \times 10^6$ CFU/mL in TSB and treated with DMSO, vancomycin (20 μg/mL, $10 \times MIC$), (*R*)-ZG197 (5 μg/mL, $10 \times MIC$), (*S*)-ZG197 (80 μg/mL, $10 \times MIC$), or ONC212 (5 μg/mL, $10 \times MIC$). After 6 h incubation at 37 °C, 220 rpm, 200 μL culture medium was taken and washed with PBS for three times. Dilutions of each condition were then plated in the absence of compounds and grown on TSA plates at 37 °C overnight. CFUs were measured by counting the number of colonies on the next day.

## MRSA persister cell assay

Two assays were performed to test effects of (*R*)- and (*S*)-ZG197 on eradicating MRSA persisters. First, *S. aureus* cells growing at stationary phase were applied[61]. After overnight culturing, stationary phase MRSA USA300 were prepared in BHI broth and incubated in the presence of (*R*)-ZG197 (10 μg/mL), (*S*)-ZG197 (80 μg/mL), ciprofloxacin (8 μg/mL), or rifampicin (0.4 μg/mL) at 37 °C and 220 rpm with aeration. At each indicated time point (0, 24, 48, and 72 h), 100 μL of culture was taken and centrifuged at $12,396 \times g$ for 2 min. Then the mixture was washed with PBS for three times. 10-fold serial dilutions were performed and 10 μL of each dilution was removed and spotted onto TSA plates. All plates were placed at 37 °C overnight. CFUs were measured by counting the number of colonies on the next day.

The second persister cell assay was performed using the biphasic killing analysis[23]. Overnight, stationary-phased USA300 culture was diluted at 1:100 in TSB. Bacteria were first incubated with ciprofloxacin (8 μg/mL, $10 \times MIC$) for 6 h at 37 °C with aeration at 220 rpm, and the remaining cells formed surviving persisters. Then compounds at $10 \times MIC$, including (*R*)-ZG197 (10 μg/mL), (*S*)-ZG197 (80 μg/mL), ADEP 4 (5 μg/mL), ciprofloxacin (8 μg/mL), or rifampicin (0.4 μg/mL), were added and 100 μL samples were taken at indicated time point (0, 2, 4, 6, 8, 12, 24, and 48 h) and centrifuged at $12,396 \times g$ for 2 min. Then samples were resuspended in PBS. CFUs were measured by counting the number of colonies on the next day.

## Cytotoxicity assay

The toxicity of the compounds on 293T/17 and HK-2 cell lines was performed by 3-(4,5-dimethylthiazol-2-yl)-2,5-diphenyltetrazolium bromide (MTT) assay. Mammalian cells (3000 cells/well) were seeded in 96-well plates allowed to adhere overnight and treated with a diluted series of ICG-001, (*S*)-ZG197, (*R*)-ZG197, or ONC212 for 72 h. Subsequently, 10 μL prepared MTT solution (5 mg/mL in PBS) was added to each well and the cells were then incubated for another 4 h at 37 °C. After removing the supernatant, the formazan precipitate in each well was dissolved in DMSO and the absorbance was measured at a wavelength of 490 nm by Tecan Spark 10 m plate reader. DMSO was used as a control and its viability was considered as 100%. $IC_{50}$ values were calculated by fitting dose–response curves. All experiments were performed in triplicate.

## Wnt/β-catenin luciferase reporter assay

Wnt/β-catenin signaling was detected using TOPFlash/FOPFlash reporter assays as previously described with modifications[39]. TOPFlash (#21-170, Millipore) was used as a Wnt-responsive reporter and FOP-Flash (#21-169, Millipore) was a mutant Wnt-irresponsive reporter. About $1.5 \times 10^5$ HEK 293 T/17 or HK-2 cells were seeded in 24-well

plates. After attachment, cells were transfected with 500 ng Firefly reporter TOPFlash or FOPFlash using Lipofectamine 2000 (Invitrogen, USA) according to manufacturer's instructions. Meanwhile, 50 ng of the Renilla reporter pRL-TK (Promega) was co-transfected as an internal control to correct for differences in both transfection and harvest efficiencies. After 6 h post-transfection, the culture medium was changed into DMEM growth medium in the presence of 20 mM LiCl and compounds at indicated concentrations. After 24 h, cells were lysed and detected by Dual-Luciferase Reporter Assay System (#E1910, Promega) according to manufacturer's instructions. The Firefly luciferase activity (TOPFlash or FOPFlash) were normalized with Renilla luciferase activity (pRL-TK). Wnt signaling activity was represented as normalized TOPFlash/FOPFlash values.

### Zebrafish infection model

Zebrafish (7 to 11 months old, $300 \pm 50$ mg, regardless of gender) were maintained in a 10-L tank with regular feeding for at least three days before subjecting to experiments. The safety profiles were conducted by intraperitoneally injecting the uninfected zebrafish ($n = 11$ per group) with 25, 50, or 100 mg/kg compounds (dissolved in 5% DMSO, 20% PEG 300, 5% solutol, and 70% PBS). For determination of an appropriate staphylococcal load for infection, zebrafish were administered with a serial CFUs of bacteria and the survival rates were monitored over 5 days. One group zebrafish was intraperitoneally injected with the same volume of PBS (8 μL) as blank control.

For investigation on anti-infective efficacy of compounds, the zebrafish were intraperitoneally injected with *S. aureus* strains (USA300, Newman, Newman Δ*clpP* or MRSA XJ049) at indicated CFUs. After 30 min post infection, zebrafish were divided randomly into different groups followed by administration of compounds at a dose of 25 or 50 mg/kg and vehicle control. Survival rates of zebrafish were recorded for 5 days.

### Murine skin infection model

The infection model was established according to previously reported method with some modifications[62]. The backs of mice (6 to 8 weeks old, 18-20 g) were shaved and treated with depilatory cream one day before infection. Overnight cultures of *S. aureus* USA300 were diluted at 1:1000 and grown to $OD_{600}$ at 0.6. Then cultures were washed by PBS three times and resuspended at a concentration of $5 \times 10^7$ CFU/mL. Before infection, mice were anesthetized with an intraperitoneal injection of 70 mg/kg pentobarbital sodium. An aliquot of 50 μL *S. aureus* USA300 suspension (containing $2.5 \times 10^6$ CFUs) were subcutaneously injected into the shaved area. At 16 h post infection, mice were randomly divided into different groups ($n = 9$ per group). The lesion area was measured and calculated using the formula $A = \pi(L/2) \times W/2$ ($A$, area; $L$, length; $W$, width). The difference of the initial lesion areas has no significance in each group. Then 50 μL of 7.5 mg/kg compounds (dissolved in 5% DMSO, 20% PEG 300, 5% solutol, and 70% PBS) and vehicle (as negative control) were subcutaneously injected near the infected skin. The treatments were applied twice daily for 3 days. Mice were then sacrificed and the skin was separated from the underlying fascia and muscle tissue. One of the nine skin samples in each group was randomly used and fixed in 4% paraformaldehyde and embedded in paraffin. Thin sections were cut and stained with H&E for microscopic observations. The remaining eight infected skin samples in each group were aseptically excised and weighed, and subsequently homogenized in 1 mL of PBS. The serially diluted homogenate was plated onto TSA plates and incubated at 37 °C overnight. The bacterial counts were expressed as $\log_{10}$ CFUs/mg of tissue.

### Statistics and reproducibility

All statistical analyses were performed using GraphPad Prism 8 (GraphPad software). Log-rank tests were performed to compare the two survival curves. Two-tailed unpaired Student's t-tests were performed to compare the means between groups. Data are presented as mean ± SD. $P$ values are indicated in the figures. The number of independent experiments/mice/samples are mentioned in the figure legends. All attempts at replication are successful. All blots and gels were performed in triplicate and a single experimental image is shown. Murine skin infection experiments, SEM and H&E staining were performed once. At least 5 representative images of SEM were recorded from each sample. Three representative images of H&E staining were taken in each sample and one is shown. All images recorded on SEM or H&E staining indicated a similar trend.

### Reporting summary

Further information on research design is available in the Nature Portfolio Reporting Summary linked to this article.

## Data availability

The atomic coordinates and structure factors data generated in this study have been deposited in the Protein Data Bank (PDB, www.pdb. org) under accession code 7WH5 for ZG180/*Hs*ClpP, 7WID for ZG180/*Sa*ClpP, 7XBZ for (*R*)-ZG197/*Sa*ClpP, and 7WGS for (*S*)-ZG197/*Sa*ClpP structure, respectively. Other X-ray structural data used in this study are available in the PDB database under accession code 6TTY, 6TTZ, 3STA and 1TG6. Amino acid sequence can be found at National Center for Biotechnology Information (NCBI) with the accession number of NP_006003 for *Hs*ClpP, CAA06443 for *Mo*ClpP, NP_001018520 for *Da*ClpP, KFL07692 for *Sa*ClpP, and CAD6014684 for *Ec*ClpP. All relevant data generated in this study are provided in the Supplementary Information and Source Data file. Source Data is provided with this paper. Data is available from the corresponding authors upon request. Source data are provided with this paper.

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

## Acknowledgements

We are grateful to Dr. Hanne Ingmer (The Royal Veterinary and Agricultural University, Denmark) for the gift of *S. aureus* NCTC 8325-4 and 8325-4/Δ*clpP* and Dr. Jiahai Zhou (Shanghai Institute of Organic Chemistry, China) for the gift of the pPSUMO vector. We thank the staff from BL19U1 and BL02U1 beamline at Shanghai Synchrotron Radiation Facility for data collection support. We thank the staff in Laboratory Animal Model Department, Shanghai Public Health Clinical Center, Fudan University for the facility support. This work was funded by National Natural Science Foundation of China (22037007, 81861138046 and 21725801 to C.-G.Y. and 22107109 to T.Z.).

## Author contributions

C.-G.Y. conceived the project and supervised the research. B.W. conducted research with the help from T.Z., P.W., Y.P., J.L., W.C., and M.Z.; T.Z. synthesized activators; Q.J., W.W., L.L., J.G., and C.-G.Y. contributed agents and data analyses; B.W., T.Z., P.W., and C.-G.Y. wrote the manuscript. All authors discussed the results and commented on the manuscript.

## Competing interests

B.W., T.Z., P.W., and C.-G.Y. are named inventors of pending patent application (CN202210377247.X, to the Chinese Patent Office) related to the work described. Patent applicant is Shanghai Institute of Materia Medica, Chinese Academy of Sciences. The status of application is official filing. The structures of ZG180, (*R*)- and (*S*)-ZG197 are covered in patent application. The remaining authors declare no competing interests.

## Ethics

The protocols of murine skin infection experiments were reviewed and approved by the Institutional Animal Care and Use Committee (IACUC) of Shanghai Public Health Clinical Center. Experiments were performed in accordance with the relevant ethical guidelines and regulations. The laboratory animal usage license (SYXK-HU-2015-0027) is certified by Shanghai Committee of Science and Technology.
