## [Peer Review File · Nature Communications]

Anti-infective therapy using species-specific activators of Staphylococcus aureus ClpPREVIEWER COMMENTS

Reviewer #1 (Remarks to the Author):

Wei et al. discovered new SaClpP agonists through structure-based design with good selectivity over HsClpP and good efficacy in vitro and in infection zebrafish model. Biochemical and biophysical data have well revealed their molecular mechanism in vitro and in cellular level. This work merits the publication in nature communications after revisions.

1. The authors screened a library using SaClpP enzymatic assay, and a hit compound ICG001 was screened out. The authors should describe why the isoleucine fragment and 4,4,4- trifluorobutyl motif were incorporated?

2. Did the authors investigate the SAR of the new identified scaffold against SaClpP? SAR study on the new scaffold may be conducive to discover more potent compounds.

3. Figure 4C, the labels of compounds are confused, please use colored labels that are consistent with the corresponding chemical structure sticks.

4. As the hit compound ICG001 is a kinase inhibitor, the author should evaluate the inhibitory activities against various kinases to assess whether the optimized compounds are off-targeted. MTT assay toward cell lines are not enough to assess their safety profiles.

5. The author should use SaClpP activator(s) as control(s) in MIC evaluation, such as ADEPs or ONC212.

6. The author said (R)-ZG197 is still positive on the mutant SaClpPI91W. The author should give more discussion on this phenomenon. The comparison of the complexes of (R)-ZG197 with SaClpPI91W and with non-mutational SaClpP maybe beneficial for revealing the mechanism of this unexpected phenomenon.

7. Why did the author use zebrafish as the model animal? Additional in vivo data on mice would be more convinced.

Reviewer #2 (Remarks to the Author):

Aberrant activation of *S. aureus* ClpP (SaClpP) could be an alternative method of discovering antibiotic drugs, while it remains difficult to develop selective SaClpP activators that can avoid disturbing the function of *Homo sapiens* ClpP (HsClpP). In this manuscript, Yang and co-workers report their design to identify (R)- and (S)- ZG197 as highly selective SaClpP activators, which increased protease activity and bound to SaClpP but not HsClpP. Subsequently, the key sites and mechanisms of (R)- and (S)- ZG197 binding to saClpP were initially revealed and their protective effects were revealed in vivo experiments. Taken all, the discoveries presented in this work provide a new starting point for developing a new therapeutic agent against MASA, which may attract considerable interest from the scientific community. The entire work is well designed and implemented. However, some data are not complete and some issues are not addressed. In this case, I would recommend a major revision to the manuscript.

1. In Supplementary Material S1B, at critical activator concentrations, all SaClpP proteolytic α -casein formed a fuzzy band, whereas all HsClpP did not. Why is there such a difference?

2. In Supplementary Material S1A, the EC₅₀ value of ADEP 4 for α -casein hydrolysis by SaClpP was 0.5±0.8 μ M. The error bar is large, please present multiple repeated data or re-evaluate.

3. In Fig 1D, the curve display of ICG001 for α -casein hydrolysis by HsClpP is incomplete, and the display range of the abscissa should be expanded to 100 μ M.

4. In Fig S1C, please list the other drugs that can activate SaClpP and calculate the hit rate. In addition, what is the basis for selecting ICG-001?

5. Please provide the process of optimization from ICG001 to ZG180, as well as the chemical formula and EC₅₀ of the other derivatives for α -casein hydrolysis by SaClpP and HsClpP, or the screening thresholds and conditions.

6. It is known from the article that the binding site of (S)ZG197 and SaClpP is I91, while the binding

sites of (R)ZG197 and SaClpP is unknown. What are your predictions for their binding sites? It is also known from the article that (R)ZG197 does not bind to HsClpP because of Tyr146, while (S)ZG197 does not bind to HsClpPPW146A, (S)ZG197 binds to HsClpPPW146 Δ C(S), and ZG197 does not bind to HsClpP because of the joint action of Tyr146 and C-terminus, failing to state is the relationship between (S)ZG197 and C-terminus dependent on Tyr146, and can the EC50 of HsClpP Δ C with (S)ZG197 be supplemented?

7. In the MIC assay, the MIC of (S)ZG197 was greater than 256 μ g/mL for the Sa ClpP I91 mutant, whereas the MIC of (R)ZG197 was 0.5. (S)ZG197 is sensitive to single point mutations, and what is the frequency of spontaneous resistance in (S)ZG197? In addition, spontaneous resistance mutants of (R)ZG197 may also have implications for elucidating the binding site of (R)ZG197 to SaClpP.

8. In the SEM assay, the authors believe that the larger cell volume is a phenomenon caused by the decrease of FtsZ, and I strongly agree with this point. In the experiment the negative control is the DMSO group, please set the positive control (the control group should be the FtsZ inhibitor treatment group that has been reported so far).

An inhibitor of FtsZ with potent and selective anti-staphylococcal activity.

TXA709, an FtsZ-targeting benzamide prodrug with improved pharmacokinetics and enhanced in vivo efficacy against methicillin-resistant *Staphylococcus aureus*.

Reviewer #3 (Remarks to the Author):

The focus of the ms is on the discovery of potent and selective molecules that activate *S. aureus* ClpP and exploring their potential utilization as a class of anti-infective agents that act by a distinct mechanism of action.

The ms has several strengths: 1) the design of the inhibitors is based on a sound biochemical rationale; 2) the investigators have identified small-molecule activators that potently and selectively inhibit SaClpP over HsClpP and, importantly, have in vitro activity against multidrug-resistant MRSA; 3) the findings are highly significant in terms of ushering a new class of anti-infectives and addressing a dire and pressing need related to multidrug-resistant MRSA infections; 4) the results are clearly presented and defended; 5) the experimental methodologies are sound and well-described; and 6) the ms is free of typos. Overall, the scientific import of the findings is high. Publication of the ms is recommended.

Reviewer #4 (Remarks to the Author):

This manuscript that describes the characterization of R and S isomers of a compound Z197 as specific inhibitors of SaClpP. The compounds were characterized structurally and biologically. The results are interesting and should have broad interest. The work was carefully done and the results significant.

Given the structures are a moderately large component of the manuscript, a clearer description should be given of the ligand binding site. A LIGPLOT+ figure detailing covalent bonds and a sequence alignment including Danio Rerio and other representative bacterial sequences would provide support to the conclusions that the C-terminal sequence is important.

In addition, ClpP is a dynamic protein that undergoes concerted conformational changes in binding substrates and inhibitors. For example, the amino terminal 20 or so residues are observed in a range of conformations. There are also conformational changes that result in compression of the molecule under certain circumstances. No analysis of changes outside of the ligand binding site was described and this represents a significant weakness that should be addressed.

Minor points

The syntheses of the new compounds should be described in detail so that it may be reproduced.

As a minor note, there are a number of typos and grammatical inconsistencies that should be corrected.

Response to the reviewers' comments

Anti-infective therapy using species-specific activators of *Staphylococcus aureus* ClpP
NCOMMS-22-14339-T

SUMMARY: We are grateful to the reviewers for the positive comments on this work, and we appreciate their thoughtful and constructive suggestions. Accordingly, we have carefully revised our manuscript and addressed all the concerns. The following is a summary of the major changes or modifications during our revision.

- 1) We have performed additional experiments to assess the safety profiles of (*R*)- and (*S*)-ZG197 in mammalian cells and on zebrafish. Also, we have evaluated the anti-infective efficacy of our species-selective activators in murine skin models.
- 2) We screened resistant mutants of (*R*)- and (*S*)-ZG197 on *SaClpP* and provided the resistance rate. Additionally, we have assayed the activity and binding affinity of (*R*)- and (*S*)-ZG197 on the C-terminal truncated *HsClpP* (*HsClpP* Δ C).
- 3) We ran LigPlot+ analysis that provides a clear description of the ligand binding site and added the sequence alignments of ClpP from other species. Moreover, a detailed discussion about the structural analysis was added in our revision.
- 4) The synthetic methods and characterization for the newly-synthesized compounds were added in Supplementary methods in the revision.
- 5) The grammar and spelling were carefully corrected and the format for citations was adjusted according to reviewer's suggestions.

The following are our point-by-point responses to the comments/suggestions from the reviewers:

1. Responses to the comments of Reviewer 1:

General Comments from Reviewer 1: Wei et al. discovered new *SaClpP* agonists through structure-based design with good selectivity over *HsClpP* and good efficacy in vitro and in infection zebrafish model. Biochemical and biophysical data have well

revealed their molecular mechanism *in vitro* and in cellular level. This work merits the publication in nature communications after revisions.

Response: We thank the reviewer for the positive comments and constructive suggestions. We have carefully revised our manuscript accordingly. Please see our point-by-point response to the specific comments below.

Comment 1 from Reviewer 1: The authors screened a library using SaClpP enzymatic assay, and a hit compound ICG001 was screened out. The authors should describe why the isoleucine fragment and 4,4,4- trifluorobutyl motif were incorporated?

Response: We thank the reviewer for this question. After the HTS, we have performed a synthetic optimization of the hit compound ICG-001 for more potent activators by using varying (unnatural) amino acids as key substituted motif, which yielded over 200 ICG-001 derivatives. Among them, ZG180, which contains the isoleucine fragment and 4,4,4-trifluorobutyl motif, displayed significantly improved activity *in vitro* and in cells compared to the hit compound ICG-001 which has the aromatic substituents on the relevant positions. Thus, we began the structure-based design of selective activators (*R*)- and (*S*)-ZG197 based on the structural complex of SaClpP bound with ZG180. The detailed synthetic methods and characterizations on these SaClpP activators have been provided in the Supplementary methods.

Comment 2 from Reviewer 1: Did the authors investigate the SAR of the new identified scaffold against SaClpP? SAR study on the new scaffold may be conducive to discover more potent compounds.

Response: Thanks for the suggestions. Yes, we investigated the SAR of this newly identified scaffold. As mentioned in previous response, we have more than 200 analogues synthesized for more potent activators, and we choose (*R*)- and (*S*)-ZG197 as tool compounds in this study. We are going to submit the SAR investigations to a more specialized journal, which would enhance the impact of this current work.

Comment 3 from Reviewer 1: Figure 4C, the labels of compounds are confused, please

use colored labels that are consistent with the corresponding chemical structure sticks.

Response: We thank the reviewer for the great suggestions. We have made changes of the color mode in Fig. 4C and checked these color labels in other figures accordingly. Now the colored labels for compounds are consistent after revision.

Comment 4 from Reviewer 1: As the hit compound ICG001 is a kinase inhibitor, the author should evaluate the inhibitory activities against various kinases to assess whether the optimized compounds are off-targeted. MTT assay toward cell lines are not enough to assess their safety profiles.

Response: We agreed with the reviewer on further assessments of compound safety profiles. To the best of our knowledge, the activity of ICG-001 as a kinase inhibitor has not been reported in literature. ICG-001 was initially established as an inhibitor for the interaction of β -catenin and CREB-binding protein (CBP) (*PNAS*, **2004**, 101, 12682-12687). To test the off-target effects of our activators, we performed a Wnt/ β -catenin Luciferase Reporter Assay to evaluate the effects of (*R*)- and (*S*)-ZG197 on Wnt signaling pathways, and we found that our activators displayed minimal effects on Wnt signaling in HEK 293T/17 and HK-2 cell lines at a high concentration of 20 μ M. The data have been added in Supplementary Fig. 6a in our revised manuscript. In addition, we further assessed the safety profiles of our compounds on zebrafish. Results indicated that (*R*)- and (*S*)-ZG197 at an administration of 100 mg/kg still exhibited excellent safety *in vivo*, while the pan-activator ONC212 significantly reduced the survival of zebrafish at a dose of 100 mg/kg, indicating excellent safety profiles of our selective SaClpP activators. We have added these new data in Supplementary Fig. 6c.

Comment 5 from Reviewer 1: The author should use SaClpP activator(s) as control(s) in MIC evaluation, such as ADEPs or ONC212.

Response: We have assayed ONC212 as a control in MIC evaluation in the revision. The new data was added in Fig. 5e.

Comment 6 from Reviewer 1: The author said (*R*)-ZG197 is still positive on the mutant

SaClpPI91W. The author should give more discussion on this phenomenon. The comparison of the complexes of (R)-ZG197 with SaClpPI91W and with non-mutational SaClpP maybe beneficial for revealing the mechanism of this unexpected phenomenon.

Response: This is a great point. As shown in Fig. 4c, when we aligned the structural complexes of (R)-, (S)-ZG197-bound SaClpP to the structure of ZG180-bound HsClpP, we found that the naphthyl motif in (S)-ZG197 exhibited direct clash to the bulky W146 in HsClpP, which might explain why (S)-ZG197 is inactive towards HsClpP. Also, in this structural alignment shown in Fig. 4c, we observed that (R)-ZG197 showed a binding pose in the hydrophobic site that is quite similar to that used by ZG180 for binding to HsClpP. Because ZG180 is active on both ClpP proteases, it suggests that the binding pocket of SaClpPI91W might be still favorable for (R)-ZG197 binding. In addition, the methyl substituent in (R)-ZG197 is still spatially away from the W91 in SaClpPI91W mutant. Therefore, it is predictable that this methyl motif in (R)-ZG197 most likely contributed hydrophobic interaction to the W91 when binding to SaClpPI91W. Although we hope we have well explained why (R)-ZG197 is still positive on the mutant SaClpPI91W, a structural complex of (R)-ZG197 bound with SaClpPI91W would clearly uncover the mechanism underlying. According to this suggestion, we have added a brief discussion on this point in our revision, “It should be noticed that (R)-ZG197 is still positive on the SaClpPI91W mutant, which is because the methyl substituent in (R)-ZG197 could contribute hydrophobic affinity with W91 in the SaClpPI91W mutant while it likely clashes with W146 in HsClpP (Fig. 4c). A resolved structure of (R)-ZG197/SaClpPI91W complex would be beneficial for revealing the mechanism of this phenomenon.”

Comment 7 from Reviewer 1: Why did the author use zebrafish as the model animal?

Additional in vivo data on mice would be more convinced.

Response: Many thanks for allowing us to clarify this concern. Zebrafish have been broadly used in drug discovery (*Nat Rev Drug Discov*, **2015**, 14, 721-31.). Moreover, the immune systems and genetic sequence are highly conserved between human and zebrafish, making it an excellent tool in antibacterial research (*Pharmaceuticals (Basel)*,

2021, 14, 594; *Trends Microbiol*, 2020, 28, 10-18). Keeping these advantages in mind, we chose zebrafish as the animal models for assessing the efficacy and safety profiles of our SaClpP activators. We agreed with the reviewer that the mice infection model will further show the *in vivo* efficacy. Accordingly, we have conducted experiments on a murine skin infection model to further assess the *in vivo* antibacterial potency of our selective activators at Shanghai Public Health Clinical Center. Lesions were treated twice over 24 h by subcutaneous injection of DMSO or our activators. Vancomycin was used as a positive control. Results indicated that both (R)- and (S)-ZG197 demonstrated significant anti-infective efficacy *in vivo*. These data could be found in Fig. 6e-h.

Reviewer #2:

General Comments from Reviewer 2: Aberrant activation of *S. aureus* ClpP (SaClpP) could be an alternative method of discovering antibiotic drugs, while it remains difficult to develop selective SaClpP activators that can avoid disturbing the function of *Homo sapiens* ClpP (HsClpP). In this manuscript, Yang and co-workers report their design to identify (R)- and (S)- ZG197 as highly selective SaClpP activators, which increased protease activity and bound to SaClpP but not HsClpP. Subsequently, the key sites and mechanisms of (R)- and (S)- ZG197 binding to saClpP were initially revealed and their protective effects were revealed in vivo experiments. Taken all, the discoveries presented in this work provide a new starting point for developing a new therapeutic agent against MASA, which may attract considerable interest from the scientific community. The entire work is well designed and implemented. However, some data are not complete and some issues are not addressed. In this case, I would recommend a major revision to the manuscript.

Response: We thank the reviewer for the positive recommendation. We have carefully revised our manuscript accordingly and provided point-by-point response to the specific comments.

Comment 1 from Reviewer 2: In Supplementary Material S1B, at critical activator concentrations, all SaClpP proteolytic α -casein formed a fuzzy band, whereas all

HsClpP did not. Why is there such a difference?

Response: We have observed the fuzzy bands in almost all of the critical concentrations (near EC₅₀) in α -casein hydrolysis by the activated *SaClpP*. We speculated that these fuzzy bands were those α -casein that was partially degraded. In addition, this fuzzy band appears at around 24 Kd in molecular weight, which are quite close to the molecular weight of *HsClpP* and most likely overlapped with the bands of *HsClpP*. Thus, this phenomenon was not observed in α -casein hydrolysis by the activated *HsClpP*. Anyway, it needs further investigations in future.

Comment 2 from Reviewer 2: In Supplementary Material S1A, the EC₅₀ value of ADEP 4 for α -casein hydrolysis by *SaClpP* was 0.5±0.8 μ M. The error bar is large, please present multiple repeated data or re-evaluate.

Response: We are sorry for making this mistake. We have performed this experiment and corrected this data in our revision in Supplementary Fig. 1a.

Comment 3 from Reviewer 2: In Fig 1D, the curve display of ICG001 for α -casein hydrolysis by *HsClpP* is incomplete, and the display range of the abscissa should be expanded to 100 μ M.

Response: According to this suggestion, we have expanded the concentration of ICG-001 to 100 μ M for completement of α -casein hydrolysis by *HsClpP*. The revised data could be found in Fig. 1d.

Comment 4 from Reviewer 2: In Fig S1C, please list the other drugs that can activate *SaClpP* and calculate the hit rate. In addition, what is the basis for selecting ICG-001?

Response: We understand this issue. However, please allow us to hold these structures for future investigations, because the other drug hits in the HTS could also serve as starting points for further development of new and potent ClpP activators. In this work, five hit compounds stood out in our HTS of a library containing 3,896 compounds, so the hit rate is about 0.13%. We have added this screening hit rate in the figure legend of Supplementary Fig. 1c. We also added a statement to clarify why ICG-001 was

selected in this work, “Considering distinct scaffold of ICG-001 and the practicable synthesis of derivatives, we chose ICG-001 as a starting point for follow-up synthetic optimization.”

Comment 5 from Reviewer 2: Please provide the process of optimization from ICG001 to ZG180, as well as the chemical formula and EC50 of the other derivatives for α -casein hydrolysis by SaClpP and HsClpP, or the screening thresholds and conditions.

Response: Thanks for this suggestion. In this study, we mainly focused on the activity and mechanism of (*R*)- and (*S*)- ZG197 that selectively promoted *SaClpP* in protein hydrolysis rather than *HsClpP*. The SAR study is a vast amount of work, which could be a separate story for a more specialized journal with a scope of medicinal chemistry. We think the follow-up publication would enhance the impact of this current work. We have provided the detailed synthetic methods and characterizations on our *SaClpP* activators in the Supplementary methods.

Comment 6 from Reviewer 2: It is known from the article that the binding site of (*S*)ZG197 and SaClpP is I91, while the binding sites of (*R*)ZG197 and SaClpP is unknown. What are your predictions for their binding sites? It is also known from the article that (*S*)ZG197 does not bind to HsClpP because of Trw146, while (*S*)ZG197 does not bind to HsClpPPW146A, (*S*)ZG197 binds to HsClpPPW146 Δ C(*S*), and ZG197 does not bind to HsClpP because of the joint action of Trw146 and C-terminus, failing to state Is the relationship between (*S*)ZG197 and C-terminus dependent on Trw146, and can the EC50 of HsClpP Δ C with (*S*)ZG197 be supplemented?

Response: We thank the reviewer for the great idea that directed us to the joint action of W146 and C-terminus in *HsClpP*. In this study, we have resolved the crystal structures of both (*R*)-ZG197 and (*S*)-ZG197 bound to *SaClpP*, which clearly showed the binding mode of the two activators toward *SaClpP*. To clearly demonstrate these interactions, we have applied LigPlot+ to display the interaction mode of our activators in the hydrophobic binding pocket of *SaClpP* (Supplementary Fig. 4b).

To investigate why our activators fail to bind to and activate *HsClpP*, we performed

structural alignment and mutagenesis study. The Fig. 4c showed that the absolute configuration of the methyl substituent on ZG197 make the two naphthyl motif take opposite direction in the binding pockets of *SaClpP*. In addition, the naphthyl motif in (*S*)-ZG197 appeared to clash with W146 in *HsClpP*, indicating that W146 is an important residue to exclude (*S*)-ZG197 binding to *HsClpP*. However, (*S*)-ZG197 failed to bind to the W146A mutant of *HsClpP*, while it bound to *HsClpPW146A* with C-terminal truncation (*HsClpPW146A* Δ C). These data indicated that the joint action of W146 and C-terminus in *HsClpP* is key to exclude (*S*)-ZG197 binding to *HsClpP*. According to the reviewer's suggestion, we supplemented the binding data and EC₅₀ of (*S*)-ZG197 with *HsClpP* Δ C and updated these data of (*S*)-ZG197 and *HsClpPW146A* Δ C in Fig. 4j and 4k. Results indicated that the relationship between (*S*)-ZG197 and *HsClpP* is dependent on both W146 and C-terminus.

The Fig. 4c also showed that the naphthyl motif in (*R*)-ZG197 positioned away from W146 of *HsClpP*, while the methyl motif might have clash with the bulky W146. Indeed, (*R*)-ZG197 is active on the mutant of *HsClpPW146A* as showed in Fig. 4g and 4h, which explained why (*R*)-ZG197 failed to bind to the wild type *HsClpP*. Interestingly, (*R*)-ZG197 is also active on *HsClpP* Δ C truncation (Fig. 4j), and the activity was further enhanced on *HsClpPW146A* Δ C. These data also indicate the joint action of W146 and C-terminus for (*R*)-ZG197 binding to *HsClpP*.

We thank the reviewer for pointing us the joint action effect of W146 and C-terminus (bearing two proline residues) in *HsClpP*, which is likely a crucial factor for discrimination of our activators binding and activation. It should be noticed that other structural motif in ClpPs and/or configuration issue of small-molecule activators might also have additional effect of the activators on selective binding to *SaClpP* rather than *HsClpP*, which needs further investigations.

Comment 7 from Reviewer 2: In the MIC assay, the MIC of (*S*)ZG197 was greater than 256 μ g/mL for the *Sa ClpP* I91 mutant, whereas the MIC of (*R*)ZG197 was 0.5. (*S*)ZG197 is sensitive to single point mutations, and what is the frequency of spontaneous resistance in (*S*)ZG197? In addition, spontaneous resistance mutants of (*R*)ZG197 may

also have implications for elucidating the binding site of (R)ZG197 to SaClpP.

Response: We thank the reviewer for the great suggestion. We have performed experiments to screen resistant mutants of (R)- and (S)-ZG197 and test the resistance rate in *S. aureus* 8325-4 strain according to the published protocol (*Chembiochem*, **2020**, *14*, 1997-2012). Our data and others indicated that the resistance rates of (R)-ZG197, (S)-ZG197, ADEP 4 and ONC212 are comparable. The frequencies of spontaneous resistance in (R)-ZG197, (S)-ZG197, ADEP 4 and ONC212 are 8.8×10^{-7} , 1.2×10^{-6} , 2×10^{-7} and 5.7×10^{-7} respectively. In addition, we observed that the sites of spontaneous resistance mutants are randomly distributed, and all of the mutants are located outside of the hydrophobic binding pocket of ClpP activators. It seems difficult to elucidate the binding site of (R)-ZG197 by screening the generation of spontaneous resistance mutants. We have added a paragraph before “**Anti-infective effects of (R)- and (S)-ZG197 in vivo**” and provided Supplementary Table 2 to clarify this point in the revision.

Comment 8 from Reviewer 2: In the SEM assay, the authors believe that the larger cell volume is a phenomenon caused by the decrease of FtsZ, and I strongly agree with this point. In the experiment the negative control is the DMSO group, please set the positive control (the control group should be the FtsZ inhibitor treatment group that has been reported so far). An inhibitor of FtsZ with potent and selective anti-staphylococcal activity. TXA709, an FtsZ-targeting benzamide prodrug with improved pharmacokinetics and enhanced in vivo efficacy against methicillin-resistant *Staphylococcus aureus*.

Response: Thank the reviewer for the great suggestion. It has been shown that the FtsZ inhibitor, TXA709, a prodrug of TXA707, indeed induced enlarged cell volume of MSSA 8325-4 under the examination of transmission electron microscopy (TEM) (*Antimicrob Agents Chemother.* **2015**, 59(8): 4845-4855). Interestingly, our activators induced a very similar phenomenon with the known results reported in literature. We have cited this reference in our revised manuscript.

Reviewer #3:

General Comments from Reviewer 3: The focus of the ms is on the discovery of potent and selective molecules that activate *S. aureus* ClpP and exploring their potential utilization as a class of anti-infective agents that act by a distinct mechanism of action. The ms has several strengths: 1) the design of the inhibitors is based on a sound biochemical rationale; 2) the investigators have identified small-molecule activators that potently and selectively inhibit SaClpP over HsClpP and, importantly, have in vitro activity against multidrug-resistant MRSA; 3) the findings are highly significant in terms of ushering a new class of anti-infectives and addressing a dire and pressing need related to multidrug-resistant MRSA infections; 4) the results are clearly presented and defended; 5) the experimental methodologies are sound and well-described; and 6) the ms is free of typos. Overall, the scientific import of the findings is high. Publication of the ms is recommended.

Response: We thank the reviewer for the recommendation for publication.

Reviewer #4:

General Comments from Reviewer 4: This manuscript that describes the characterization of R and S isomers of a compound Z197 as specific inhibitors of SaClpP. The compounds were characterized structurally and biologically. The results are interesting and should have broad interest. The work was carefully done and the results significant.

Response: We thank the reviewer for the positive comments. We have carefully revised our manuscript accordingly.

Comment 1 from Reviewer 4: Given the structures are a moderately large component of the manuscript, a clearer description should be given of the ligand binding site. A LIGPLOT+ figure detailing covalent bonds and a sequence alignment including Danio Rerio and other representative bacterial sequences would provide support to the conclusions that the C-terminal sequence is important.

Response: According to this great suggestion, we have used the LigPlot+ to show the interaction mode of our activators with SaClpP in the hydrophobic binding pocket. In

addition, we performed sequence alignment of ClpPs from *Mus Musculus*, *Homo Sapiens*, *S. aureus NCTC 8325*, *Danio Rerio*, and *E. coli* (Supplementary Fig. 2a). Results indicated that the lid motif of P248 and P249 residues in *HsClpP* are also conserved in other eukaryotic ClpPs, like *MoClpP* and *DaClpP*, while the C-terminal motif is shortened in bacterial ClpPs and the corresponding lid motif is absent, likely indicating an important role of the C-terminus for activator recognition. However, the biochemistry in depth and molecular mechanism of the unique C-terminal motif in eukaryotic ClpPs still needs more investigations. We have showed more analysis on ligand binding in Supplementary Fig. 4b and added more descriptions on structural features in the revision.

Comment 2 from Reviewer 4: In addition, ClpP is a dynamic protein that undergoes concerted conformational changes in binding substrates and inhibitors. For example, the amino terminal 20 or so residues are observed in a range of conformations. There are also conformational changes that result in compression of the molecule under certain circumstances. No analysis of changes outside of the ligand binding site was described and this represents a significant weakness that should be addressed.

Response: Thanks for this suggestion. We have performed structural alignments of the reported structural complexes. Results indicated that these crystal structures of *SaClpP* binding with ADEP 4, (R)- and (S)-ZG197 shared highly similar extended conformation. No significant differences could be observed in the catalytic triad. In addition, only four or five out of the fourteen N-terminal loops are ordered in the structural complexes of (R)- and (S)-ZG197/*SaClpP*, implying a highly dynamic binding mode for our activators. These structural analyses have been added in Supplementary Fig. 4c and 4d, and we have described structural analysis of changes outside of the ligand binding site in the revision.

Comment 3 from Reviewer 4: The syntheses of the new compounds should be described in detail so that it may be reproduced.

Response: We agreed. We have provided the detailed synthetic methods and

characterizations of ZG180, (*R*)- and (*S*)-ZG197 in the Supplementary methods.

Comment 4 from Reviewer 4: As a minor note, there are a number of typos and grammatical inconsistencies that should be corrected.

Response: We have carefully checked and corrected the typos and inconsistent grammar.

REVIEWER COMMENTS

Reviewer #1 (Remarks to the Author):

The authors addressed all the issues raised by this reviewer and I agree to publish the revised version as it is.

Reviewer #2 (Remarks to the Author):

The author's response was quite serious and logically complete. All issues have been appropriately reply, and can be accepted.

Reviewer #4 (Remarks to the Author):

This is a much-improved manuscript that addresses the major concerns of the earlier review. However, inspection of Supplementary table 1 reveal some clear issues with the reported data that must be addressed prior to acceptance.

For example, in all cases, the number of unique reflections reported for data collection and data refinement are different; and in some cases very different (7WID >23000 fewer used in refinement; 7XBZ >143000 fewer). More confusingly, lower resolution data was used in refinement (7XBZ; lower resolution limit upto 144 Å) than was collected (upto 30Å). These are very concerning. What was your reasoning for refining against less data than you collected? What is your data cut-off level?

Why are there a different number of ligand atoms in the R and S isomers of ZG197? What is the RMSD for monomers superimposed? What is the average B for protein, ligand and water molecules separately? Since this is a highly symmetrized molecule, what strategy did you use to select your free reflections and how many did you select? Why are the geometric restraints so tight in one structure relative to the others?

Since, for crystal structures, seeing is believing, and there are well-documented artefacts that can arise when data is not handled carefully, it is important for the data to be correctly reported and the refinement accurately described to be confident in the electron density in the figures. Relaxation of NCS restraints in the structures of ClpP led to the visualization of different conformations in the amino terminal 20 amino acids, which had functional consequences underlining the importance of careful structure refinement.

Response to the reviewers' comments

Anti-infective therapy using species-specific activators of *Staphylococcus aureus* ClpP
NCOMMS-22-14339A

SUMMARY: We are grateful to the reviewers for the positive recommendations on our work, and we thank the reviewer for careful checking on our structure data. The following is a summary of major changes or modifications during our revision.

We have carefully checked our data statistics and performed additional structural refinements to fix all issues raised in this review. Accordingly, the typos in Supplementary Table 1 have been corrected. In addition, we have revised Fig. 1a-b by removing out the irrelevant *HsClpP*'s activator D9 and our redundant characterization on ADEP 4 and ONC212 as the global activators in PAGE assay, which have already been described in introduction and the references have been cited. The context of discussion on D9 and the content of Supplementary Fig 1 were revised accordingly in our revision.

Reviewer #1:

Comments from Reviewer 1: The authors addressed all the issues raised by this reviewer and I agree to publish the revised version as it is.

Response: Many thanks for the recommendation.

Reviewer #2:

Comments from Reviewer 2: The author's response was quite serious and logically complete. All issues have been appropriately replied and can be accepted.

Response: We thank the reviewer for the recommendation.

Reviewer #4:

Comments from Reviewer 4: This is a much-improved manuscript that addresses the major concerns of the earlier review. However, inspection of Supplementary table 1

reveal some clear issues with the reported data that must be addressed prior to acceptance.

Response: We thank the reviewer for the positive comments. We have carefully revised our manuscript accordingly.

Comment 1 from Reviewer 4: For example, in all cases, the number of unique reflections reported for data collection and data refinement are different; and in some cases, very different (7WID >23000 fewer used in refinement; 7XBZ >143000 fewer). More confusingly, lower resolution data was used in refinement (7XBZ; lower resolution limit up to 144 Å) than was collected (up to 30 Å). These are very concerning. What was your reasoning for refining against less data than you collected? What is your data cut-off level?

Response: We are grateful to the reviewer for careful checking on our reports of structure data. We apologize for making mistakes on reporting our structural statistics. We have carefully checked the data sets and fixed all issues. As shown in the revised Supplementary Table 1, the correct number of unique reflections is 171564 for the 7XBZ structure, and the resolution range for the refinement is 30.0-2.15 Å that is consistent with the data collection. We have further refined our 7XBZ structure in Refmac5, and the updated $R_{\text{work}}/R_{\text{free}}$ and RMSD values have been included in Supplementary Table 1, and the new validation report is also uploaded for review.

We thank the reviewer for pointing out the difference in the numbers of reflections for data collection and refinement. Puzzled by these observations, we checked the original diffraction data sets and noticed that each data set has many diffractions with negative intensity (Please see below for some examples in the 7WID structure). Although we did not apply any data cut-off restraint during the refinement, the diffractions with negative intensity would be automatically rejected by the refinement programs of Refmac5. This explains why fewer unique reflections were used in refinement.

h	k	l	I	sigmaI
0	0	76	-2.4	3.9

0	1	76	-2.1	2.7
0	2	73	-0.2	1.2
0	7	75	-0.8	1.6
0	15	71	-1.6	1.4
.....				

In addition, we also checked several reports on data collection and refinement statistics in literatures. Similarly, there are also slightly difference in the numbers of reflections for data collection and refinement in several reported structures (please see below for the detailed information of the reported structures), while the subtle difference in the numbers of the diffractions could have no impact on the quality of the structures.

- 1) Structural insights into the mechanism of adaptive ribosomal modification by *Pseudomonas* RimK, *Proteins*, 2022 Sep 22. doi: 10.1002/prot.26429. Online ahead of print.
- 2) De novo design of discrete, stable 310-helix peptide assemblies *Nature*, 2022 Jul;607(7918):387-392.

Comment 2 from Reviewer 4: Why are there a different number of ligand atoms in the R and S isomers of ZG197? What is the RMSD for monomers superimposed? What is the average B for protein, ligand, and water molecules separately? Since this is a highly symmetrized molecule, what strategy did you use to select your free reflections and how many did you select? Why are the geometric restraints so tight in one structure relative to the others?

Response: Thanks for these great suggestions. The different number of ligand atoms is due to the different number of ligands (*R* and *S* isomers of ZG197) included in the two structures. In the (*S*)-ZG197/*Sa*ClpP complex structure, fourteen (*S*)-ZG197 activators were captured in the fourteen hydrophobic sites of the *Sa*ClpP tetradecamer. However, twelve (*R*)-ZG197 activators with strong density maps were captured in the (*R*)-ZG197/*Sa*ClpP complex structure. Due to weak electron density maps, two (*R*)-ZG197 activators were not included in the finally refined (*R*)-ZG197/*Sa*ClpP structure. We have added this statement in our revision to clarify why the number of ligand atoms in (*R*)- and (*S*)-ZG197 isomers is different. The structural alignment yields low C α root mean square deviation (RMSD) values around 0.2 Å for the monomers, and we have added this statement in figure legend of Supplementary Figure 4c in the revision.

As suggested by the reviewer, we have calculated the average B value for protein, ligand, and water molecules separately, which have been provided in Supplementary Table 1 in the revision. Compared to protein and water molecules in the (*R*)-ZG197/*Sa*ClpP complex structure, the average B value for twelve (*R*)-ZG197 activators is slightly higher. However, some (*R*)-ZG197 activators also have comparable average B values to that of proteins as observed in chain a (B value, 49.63) and chain b (B value, 38.75).

It should be noticed that we did not manually select free reflections in structure refinement. For all structures, 5% data were randomly selected by the `import_scaled` program embedded in the `ccp4i` suite as the free reflections. We have added a statement of “The free reflections were automatically selected and applied in all refinements.” in Materials and methods section “Crystallization and structure determination”.

We thank the reviewer for reminding us about the geometric restraint of the structures and allowing us to make a clarification on this concern. The 7WID, 7XBZ and 7WGS structures were refined using Refmac5 program embedded in the `ccp4i` suite, while the 7WH5 structure was refined using the `phenix.refine` program embedded in the `phenix` suite. To address whether refinement program would affect the final geometric restraint statistics, we also performed refinement of the 7WH5 structure in Refmac5 program. We found that the geometric restraints for the newly refined 7WH5 structure, including RMSD bond length (0.012) and bond angle (1.644) are comparable to all other three structures. Compared to protein and water molecules in the 7WH5 complex structure, the average B value for the ligands is slightly higher. However, the average B values for several ligands are still comparable to that of protein, including chain h (B value, 54.69) and chain i (B value, 48.13). We decide to report the Refmac5-refined 7WH5 structure in the revision, and the updated data have been included in the Supplementary Table 1, and the new version of validation report is also uploaded for review.

Comment 3 from Reviewer 4: Since, for crystal structures, seeing is believing, and there are well-documented artefacts that can arise when data is not handled carefully, it is important for the data to be correctly reported and the refinement accurately described to be confident in the electron density in the figures. Relaxation of NCS restraints in the structures of ClpP led to the visualization of different conformations in the amino terminal 20 amino acids, which had functional consequences underlining the importance of careful structure refinement.

Response: We strongly agreed with the reviewer that artefacts can arise when data was not handled carefully. To prevent any artefacts from data handling, we have double-checked all our four structures in this revision to make sure that the data were carefully handled and correctly reported, and the refinements were accurately described. During refinement, we did apply the NCS restraint. The restraints were generated by the Refmac5 program automatically. Instead of the whole chain, it was based on the local region of the protein monomer. The conformation of the N-terminal 20 amino acids is usually flexible, playing important roles to regulate ClpP. So, during the structural refinement, we pay special attention on tracing these motifs. Similar to those reported ClpP structures, our solved structural complexes of ClpP/activators have showed that some amino terminals had weak traceable electron density, while others were well folded. These observations indicated that the NCS restraints were not strictly applied to the N-terminal 20 amino acids. We then checked the molecular packing of ClpP in the crystal lattice. The well folded amino terminal amino acids are mainly stabilized by their interactions with symmetry-related ClpP molecule in the crystal lattice. We agree with the reviewer that it is a great idea to pay more attention on refinement of ClpP' N-terminal 20 amino acid conformations by carefully applying relaxation of NCS restraints.

REVIEWERS' COMMENTS

Reviewer #4 (Remarks to the Author):

This is a revised manuscript in which the authors have taken a more careful look at their data collection and refinement protocols. I am satisfied that the additional work they have done and data provided gives the informed reader sufficient information to evaluate the structures and data.

Response to the reviewers' comments

Anti-infective therapy using species-specific activators of *Staphylococcus aureus* ClpP
NCOMMS-22-14339-B

Reviewer #4:

Comments from Reviewer 4: This is a revised manuscript in which the authors have taken a more careful look at their data collection and refinement protocols. I am satisfied that the additional work they have done and data provided gives the informed reader sufficient information to evaluate the structures and data.

Response: We thank the reviewer for the recommendation.